# An *ARF* gene mutation creates flint kernel architecture in dent maize

Haihai Wang [1,8], Yongcai Huang [1,2,8], Yujie Li[1,3,8], Yahui Cui[1], Xiaoli Xiang [1], Yidong Zhu[1,3], Qiong Wang[1], Xiaoqing Wang[4], Guangjin Ma[1,3], Qiao Xiao[1,3], Xing Huang[1,3], Xiaoyan Gao[1], Jiechen Wang[1], Xiaoduo Lu[5], Brian A. Larkins[6], Wenqin Wang [7] ✉ & Yongrui Wu [1] ✉

Dent and flint kernel architectures are important characteristics that affect the physical properties of maize kernels and their grain end uses. The genes controlling these traits are unknown, so it is difficult to combine the advantageous kernel traits of both. We found mutation of *ARFTF17* in a dent genetic background reduces IAA content in the seed pericarp, creating a flint-like kernel phenotype. *ARFTF17* is highly expressed in the pericarp and encodes a protein that interacts with and inhibits MYB40, a transcription factor with the dual functions of repressing *PIN1* expression and transactivating genes for flavonoid biosynthesis. Enhanced flavonoid biosynthesis could reduce the metabolic flux responsible for auxin biosynthesis. The decreased IAA content of the dent pericarp appears to reduce cell division and expansion, creating a shorter, denser kernel. Introgression of the *ARFTF17* mutation into dent inbreds and hybrids improved their kernel texture, integrity, and desiccation, without affecting yield.

Maize (*Zea mays* subsp. *mays*) was domesticated from its wild-type progenitor, teosinte (*Zea mays* subsp. *parviglumis*), 10,000 years ago. Among the fortuitous mutations utilized by Native American plant breeders was *teosinte glume architecture1*[1], which prevented the cupule and glume from surrounding the embryo and endosperm in a hard fruit case[2]. As a consequence, the maize seed, a caryopsis, is covered by two tightly fused maternal tissues, namely the seed coat and the fruit coat[3]. The seed coat, also known as the testa, originates from the ovule integument and is a single cell layer, while the fruit coat, commonly called the pericarp, develops from the ovary wall and consists of multiple cell layers. The soft seed coat of the maize kernel greatly facilitated processing of the grain to access the starch- and protein-rich endosperm for food preparation.

As maize spread through the Americas, flints and dents were among several distinct types selected. Dent germplasms were introduced to the United States from Mexico, while flint germplasms originated from Native American populations of Southwestern America and the Great Plains[4,5]. Today, these two germplasms are widely used for breeding, but they have distinct geographical adaption. Because of their early maturity and cold tolerance traits, flint germplasms are predominantly grown at high latitudes, such as the Northern regions of the U.S. and Canada, and they are the major progenitors of Northern European maize[6,7]. Due to their ready adaptation to cold climates and rapid seed desiccation when maturing, flint hybrids are also popular in Heilongjiang, the northern most province of China.

---

[1]National Key Laboratory of Plant Molecular Genetics, CAS Center for Excellence in Molecular Plant Sciences, Shanghai Institute of Plant Physiology & Ecology, Chinese Academy of Sciences, Shanghai 200032, China. [2]State Key Laboratory of Crop Gene Exploration and Utilization in Southwest China, Sichuan Agricultural University, Chengdu 611130, China. [3]University of the Chinese Academy of Sciences, Beijing 100049, China. [4]Forestry and Pomology Research Institute, Shanghai Academy of Agriculture Sciences, Shanghai 201403, China. [5]Institute of Molecular Breeding for Maize, Qilu Normal University, Jinan 250200, China. [6]School of Plant Sciences, University of Arizona, Tucson, AZ 85721, USA. [7]Shanghai Key Laboratory of Plant Molecular Sciences, College of Life Sciences, Shanghai Normal University, Shanghai 200234, China. [8]These authors contributed equally: Haihai Wang, Yongcai Huang, Yujie Li. ✉e-mail: wang2021@shnu.edu.cn; yrwu@cemps.ac.cn

While there are many genetic differences between flint and dent germplasms, they are commonly distinguished based on their kernel architectures[6]. Dent kernels are usually longer than flint kernels, a trait positively related with kernel weight and yield. The top of dent-type kernels begins to collapse late in development, creating a soft, concave crown at maturity that is subject to breaking during grain harvesting and processing. In contrast, the top of flint kernels is convex, and the mature endosperm it contains is dense and vitreous, creating a harder kernel more resistant to shattering during harvesting[3]. However, flint types tend to be lower yielding than dents[8], which could be a consequence of a shorter kernel length or the environments in which they are grown. Dents generally desiccate more slowly than flints during maturation, leading to a higher moisture content at maturity[9]. Therefore, dent hybrids often require supplemental drying to prevent the grain from molding[10].

Grain end uses are a consideration for growing flints and dents. Vitreous endosperm is important for dry milling because it produces a greater yield of "grits", which are needed to make corn flakes and corn meal. Dent kernels, being softer, are better suited for wet milling, where starch recovery is the primary goal[11-13]. Breeders generally select for the most suitable dent and flint architectures that do not sacrifice yield.

The genetic and biochemical mechanisms responsible for dent and flint kernel traits are poorly understood[14]. Utilization of high-quality genome sequences, coupled with gene expression analysis, represents a powerful approach to investigate the genetic basis of dent and flint maize[14]. To date, no major quantitative trait locus (QTL) responsible for these kernel phenotypes has been identified; thus it is not feasible to use marker-based genetic selection to introgress flint advantages into dent germplasm, or vice versa. Because the dent phenotype is dominant, we reasoned it would be possible to identify genes associated with flint traits through mutagenesis. Accordingly, we used EMS treatment of B73, a well-known dent inbred with a well-characterized genome sequence, to screen for mutants with flint-like kernels. This screen led to the identification of a mutant showing flint kernel architecture. We designated the mutant *flint kernel architecture 1-1, or fka1-1*. The mutation results from a defective AUXIN RESPONSE FACTOR (*ARF*) transcription factor, *ARFTF17*, that affects cell length, cell number and IAA content during kernel pericarp development. ARFTF17 cannot bind auxin response elements and requires an inter-acting protein, MYB40, to regulate downstream genes, such as the auxin transporter gene, *PIN1*. Null mutation of *ARFTF17* or over-expression of *MYB40* reduces auxin accumulation and affects pericarp development, creating a flint kernel architecture. Introgression of *fka1* into several different dent inbreds and hybrids created a flint kernel phenotype without affecting important agronomic traits, including yield. The kernels manifest flint phenotypic advantages, with a convex crown, denser and more vitreous endosperm, resistance to kernel breakage, lower kernel moisture content and higher test weight. In this work, we identify a key gene *ARFTF17* that regulates maize pericarp development. Mutation of *ARFTF17* improves unfavorable traits of maize with dent shape kernels, thus holding important translational value for maize breeding.

## Results

### Cellular basis of B73 and *fka1-1* kernel phenotypes

Because the B73 inbred has the dominant dent kernel phenotype and a well characterized genome sequence (Fig. 1a), we selected it to gain insight into the basis of dent and flint kernel phenotypes. We screened a large number of $M_3$ ears from an EMS-induced B73 mutant collection and identified one with uniformly flint-like kernels (Fig. 1a). Because the mutation created a flint kernel architecture in dent maize, we designated it *flint kernel architecture 1-1*, or *fka1-1*.

B73 kernels typically have a dented, collapsed starchy crown, with most of the hard, vitreous endosperm located on the abgerminal

kernel side. In contrast, the top of *fka1-1* kernels is convex and vitreous endosperm forms on both the abgerminal and adgerminal sides of the kernel (Fig. 1b). We measured the degree of dent in the crown (as indicated in Fig. 1b), and found the values on average for B73 and *fka1-1* were + 41.50 and −3.33 degrees, respectively (Fig. 1c). *fka1-1* kernels contain 157% more vitreous endosperm than B73 kernels (Fig. 1d) and thus are more resistant to breakage (Fig. 1e). The crown of the B73 kernel starts to flatten around 15 to 25 days after pollination (DAP), collapses around 30 DAP and by 35 DAP a dent is clearly evident (Fig. 1f). The dome-shaped crown of *fka1-1* kernels is evident at 15 DAP and is maintained throughout grain filling and seed maturation (Fig. 1f). At every developmental stage, the length of the pericarp of *fka1-1* kernels, measured from the abgerminal bottom to the crown and down to the adgerminal bottom (Fig. 1f), was shorter than that of B73 kernels (Fig. 1g). The length of *fka1-1* kernels during grain filling and at maturity was always shorter than that of B73 (Supplementary Fig. 1a), while their width was only slightly and variably affected (Supplementary Fig. 1b). Despite the reduction in kernel length, there were not significant differences in 100-kernel dry weight during grain filling or at seed maturity (Fig. 1h and Supplementary Fig. 1c). In addition, *fka1-1* kernels showed a lower moisture content and a higher test weight (Supplementary Fig. 1d, e). Total starch content and total protein content (based on whole dry seed flour) were not affected by *fka1-1* (Supplementary Fig. 1f, g). Besides the data from a field in Sanya (18.2°N, 109.3°E; Fig. 1a–h and Supplementary Fig. 1a–e), similar results were obtained from a field in Shanghai (30.5°N, 121.1°E) for the kernel architecture phenotypes, 100-kernel dry weight, moisture content and test weight (Supplementary Fig. 1h–s).

To investigate the cellular differences related to the dent and flint-like crown phenotypes of B73 and *fka1-1*, we examined longitudinal sections of kernel crowns. Semi-thin sectioning at 20 DAP revealed cells of the pericarp and endosperm in B73 and *fka1-1* were firmly attached. However, starchy endosperm cells in *fka1-1* had a polygonal shape, a dense cytoplasm, and were filled with starch granules, whereas those in B73 were irregular with markedly fewer starch granules (Supplementary Fig. 1t). Transmission electron microscopy (TEM) results revealed *fka1-1* endosperm cells in the crown region contained many more starch granules and protein bodies than B73 (Supplementary. 1t). At 30 DAP, due to insufficient grain filling and desiccation, endosperm cells in the crown region of B73 were deformed and had begun to shrink from the aleurone layer inward (Fig. 1f, i). Perhaps because the starchy endosperm cells and pericarp did not collapse at the same rate in B73, the aleurone became detached from the pericarp. In contrast, the starch-filled endosperm cells in the crown of *fka1-1* maintained their polygonal shape and the convex crown (Fig. 1i).

Accumulation of starch and protein in the endosperm is usually controlled by the filial (endosperm and embryo) genotype. When reciprocal crosses were made between B73 and *fka1-1*, we found the dent phenotype occurred only when B73 was the ear parent, and vice versa for the flint-like kernel phenotype (Supplementary Fig. 1u, v). The degree of dent, kernel length, kernel width and 100-kernel dry weight of progeny seeds were not affected by the pollen genotype (Supplementary Fig. 1w–z). These results confirmed that both kernel traits are determined by the maternal rather than filial genotype.

### *FKA1* encodes the ARF transcription factor ARFTF17

To clone the gene responsible for the *fka1-1* mutation, we created an $F_2$ population from a cross with wildtype B73 (Supplementary Fig. 2a). Because kernel crown traits are maternally controlled, $F_2$ seeds produced from self-pollinated $F_1$ plants were uniformly dented, implying the dent trait is dominant (Supplementary Fig. 2a). We planted the $F_2$ seeds and sampled a piece of leaf for DNA extraction from the resulting plants. The $F_3$ seeds on $F_2$ ears were either all dent or all flint, and the dent and flint ears segregated at approximately a 3:1 ratio (dent: flint = 392: 110, $\chi2 = 2.79$; Supplementary Fig. 2a), indicating the flint

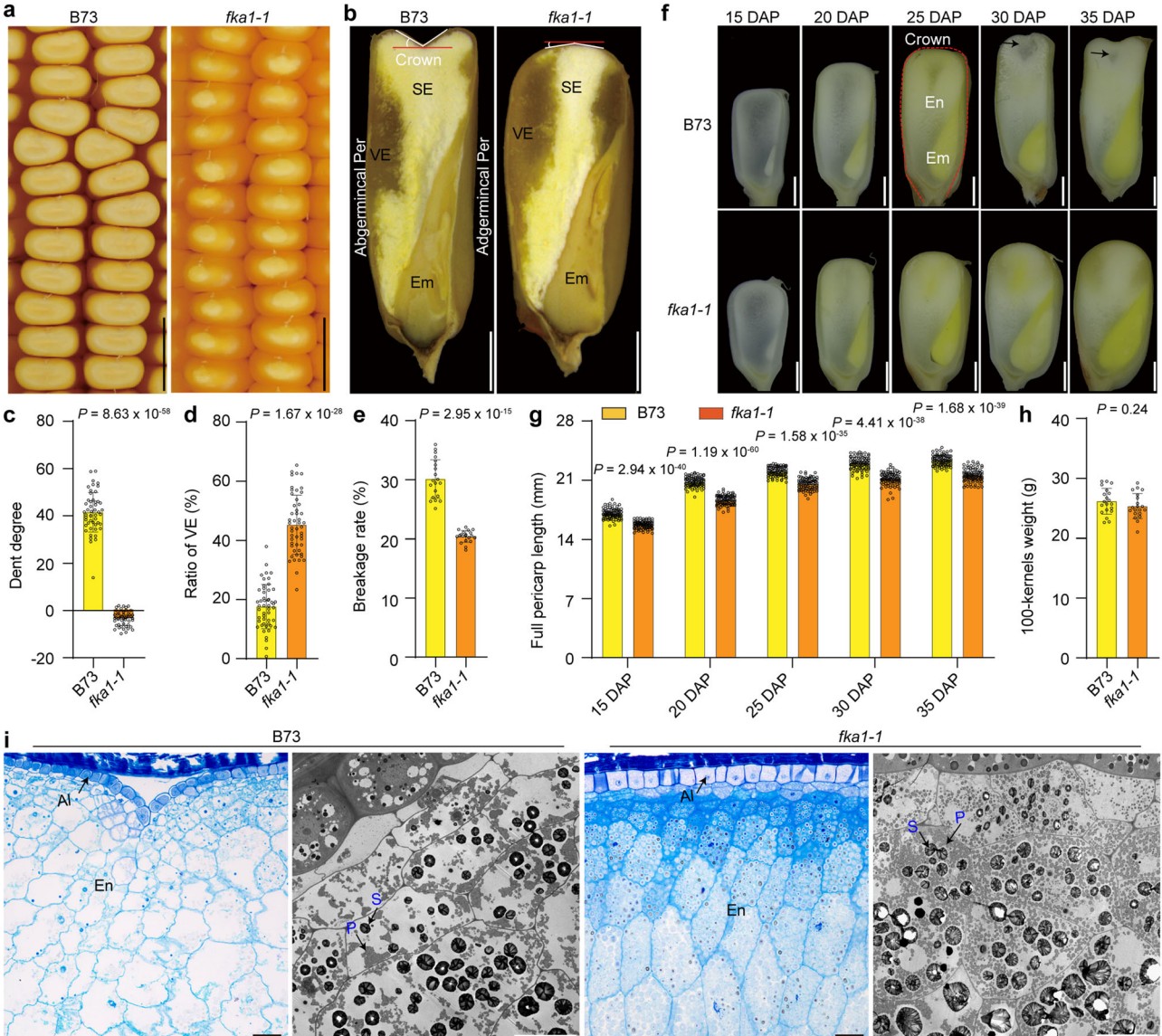

**Fig. 1 | Kernel phenotypes of B73 and *fka1-1* harvested in Sanya. a** The kernel phenotype of B73 and *fka1-1*. Scale bar, 1 cm. **b** Longitudinal-sections of mature B73 and *fka1-1* kernels. The angles of the dent on B73 and *fka1-1* kernel crowns are a measure of the degree of the dent. Scale bar, 2 mm. **c** Statistical measurements of kernel crown collapse in B73 and *fka1-1*. Schematic diagrams showing area measured are indicated in (**b**). Data are mean ± s.d. (*n* = 50 kernels from 20 ears). **d** Ratio of vitreous endosperm (VE) area to the total endosperm. Data are mean ± s.d. (*n* = 50 kernels from 20 ears). **e** Breakage rate of kernels. Data are mean ± s.d. (*n* = 20 ears). **f** Longitudinal-sections of B73 and *fka1-1* developing kernels at 15, 20, 25, 30 and 35 days after pollination (DAP). The red dotted line marks the area measured for pericarp length. Black arrows indicate area of incomplete grain filling. Scale bar, 2 mm. **g** Pericarp length of B73 and *fka1-1* kernels at different developmental time points. The schematic diagrams show measurements in (**f**). Data are mean ± s.d. (*n* = 88 kernels from 11 ears). **h** Dry weight of 100 mature kernels of B73 and *fka1-1*. Data are mean ± s.d. (*n* = 20 ears). **i** Light microscopy and TEM of sections of kernel crown regions of B73 and *fka1-1* at 30 DAP. Scale bars in semi-thin sections, 100 μm; scale bars in TEM, 20 μm. Al, aleurone; En, endosperm; Em, embryo; P, protein body; S, starch granule; SE, starchy endosperm; VE, vitreous endosperm. Two-tailed Student's *t*-tests were used to determine *P*-values shown in the (**c**–**e**, **g**, **h**).

phenotype of the mutant resulted from a single recessive gene. Based on the ear phenotypes of the F$_{2:3}$, we separately pooled all the dent and all the flint leaf DNA samples of F$_2$ plants for Bulked Segregant Analysis (BSA) sequencing. The results revealed a single peak spanning a 2-Mb region (27–29 Mb) on chromosome 5 (Supplementary Fig. 2b–d). Based on the B73 genome annotation, we identified 38 protein-coding genes of which only Zm00001d014013 harbored a single nucleotide polymorphism. This was in the gene's first exon and resulted in a C to T transition in the mutant gene relative to the B73 wild-type gene (Fig. 2a). This mutation caused the tryptophan at amino acid 332 to be replaced with a stop codon, resulting in early termination of translation.

Zm00001d014013 was predicted to encode an ARF transcription factor (TF), ARFTF17. We examined the expression of *ARFTF17* in different maize tissues by reverse transcription quantitative PCR (RT-qPCR) and found the highest transcript abundance was in the pericarp of developing seeds (Fig. 2b). We created transgenic maize lines in which the green fluorescent protein (GFP) gene was expressed with the *ARFTF17* promoter. The *ARFTF17Pro:GFP*-expressing plants showed GFP signals were primarily detected in the pericarp and pedicel region (Fig. 2c and Supplementary Fig. 2e-i). Consistent with this, between 8 and 12 DAP the ARFTF17 protein accumulated at higher levels in the pericarp than the endosperm (Fig. 2d). The transcript level of *ARFTF17* in developing pericarp of the mutant was dramatically reduced

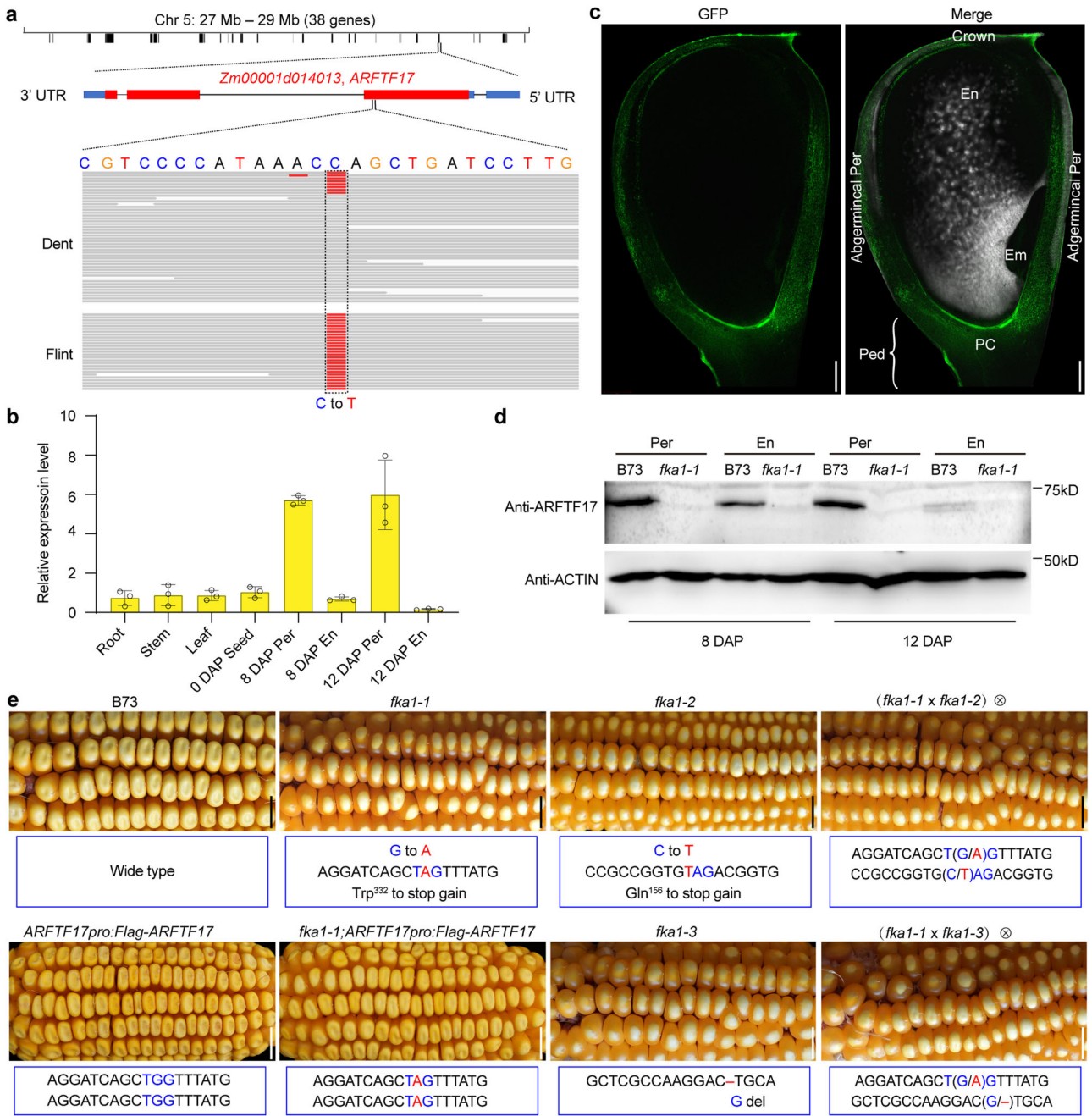

**Fig. 2 | Gene cloning and genetic verification of *fka1-1*. a** Mapping-by-sequencing of *fka1-1*. The *fka1-1* mapping interval contains the *ARFTF17* gene. Genomic sequencing reads showed a C to T mutation in the first exon of *ARFTF17* in *fka1-1*. **b** RT-qPCR analysis of *ARFTF17* expression in different tissues. The expression levels were normalized to that of *SGT1* (*Zm00001d044172*). Data are mean ± s.d. (*n* = 3 biologically independent samples). **c** GFP signal of *ARFTF17* expression in 12 DAP kernels of the transgenic maize line, *ARFTF17pro:GFP*. Scale bar, 500 µm. **d** Immunoblot analysis of ARFTF17 accumulation in the pericarp and endosperm of B73 and *fka1-1* at 8 and 12 DAP. ACTIN was used as an internal control. **e** Genetic confirmation of *fka1-1* by complementation of *fka1-1* with other alleles and *ARFTF17pro:Flag-ARFTF17*. At each mutation site, blue letters indicate wild-type nucleotides, red letters indicate the mutation, and a red dash indicates missing nucleotides. Scale bar, 1 cm. Em, embryo; En, endosperm; PC, placenta-chalaza tissue; Ped, pedicel; Per, pericarp.

(Supplementary Fig. 2j) and the protein was not evident (Fig. 2d), consistent with mutation creating a null allele.

To genetically validate that mutation of *ARFTF17* was responsible for the mutant phenotype, we obtained a different EMS-generated B73 allele from the MEMD database (http://elabcaas.cn/memd/public/index.html#/)[15], and designated it *fka1-2*. The *fka1-2* allele also contains a C to T transition in the 1st exon of the gene, replacing glycine at amino acid 156 with a stop codon (Fig.2e). The transcript level of *ARFTF17* in *fka1-2* was also greatly reduced (Supplementary Fig. 2k). A cross between *fka1-1* and *fka1-2* revealed the dent phenotype was not

recovered in F₂ seeds (Fig. 2e and Supplementary Fig. 2l–n), confirming Zm00001d014013 (*ARFTF17*) is the gene responsible for the *fka1* phenotype.

To further confirm this result, we created transgenic plants expressing a fusion gene, *Flag-ARFTF17*, driven by the *ARFTF17* promoter (Supplementary Fig. 2o). When crossed with *fka1-1* for a complementation test, the genotype of the F₂ seedlings identified the plants homozygous for *fka1-1* and containing the *ARFTF17pro:Flag-ARFTF17* transgene. RT-qPCR and immunoblot analysis of developing seeds from self-pollinated plants revealed FLAG-ARFTF17 accumulated

at high levels in the pericarp (Supplementary Fig. 2p, q). Kernels on these ears showed a dent phenotype at maturity, confirming the *fka1-1* phenotype was complemented by the transgene (Fig. 2e and Supplementary Fig. 2l–n).

## Role of *ARFTF17*

The maize genome has 33 expressed ARF genes[16] that are classified into three clades (A, B and C) based on amino acid identity with *Arabidopsis* ARF orthologs[17]. *ARFTF17* belongs to clade C with three other members, including *ARFTF2* (Zm00001d032683), *ARFTF19* (Zm00001d014507) and *ARFTF21* (Zm00001d000358)[17]. To investigate whether these four *ARFTF* genes have functional redundancy, we used genome editing via clustered regularly interspaced short palindromic repeats (CRISPR)/ CRISPR-associated protein 9 (Cas9) to create mutations and knock out their expression in the maize "Hi-II" hybrid (Supplementary Fig. 3a, b). Both A and B inbred lines used to create Hi-II have a dent phenotype. In contrast to the typical Hi-II phenotype (Supplementary Fig. 3b), ears of the transgenic plant with the CRISPR vector showed uniformly flint-like kernels (Supplementary Fig. 3c).

Because the CRISPR guide RNA was designed to target a conserved sequence in all four genes (Supplementary Fig. 3a), we could recover four single, six double, four triple and one quadruple mutant. We introgressed these mutations into B73 by backcrossing for four generations and then self-pollinated for two generations. PCR and sequencing (with primers shown in Supplementary Fig. 3a) were used to ensure seeds selected for planting each generation contained the specific gene mutations. Analysis of their progeny showed only *ARFTF17-crispr* (named as *fka1-3*), or combinations containing this mutation (*ARFTF2-cr;17-cr, ARFTF17-cr;19-cr, ARFTF17-cr;21-cr, ARFTF2-cr;17-cr;19-cr, ARFTF2-cr;17-cr;21-cr, ARFTF17-cr;19-cr;21-cr, ARFTF2-cr;17-cr;19-cr;21-cr*), displayed the flint-like kernel phenotype, while other combinations, including the triple mutant of *ARFTF2-cr;19-cr;21-cr*, exhibited a dent phenotype (Supplementary Fig. 3d). These results confirmed ARFTF17 is the major clade member, if not the only one, that regulates the dent and flint-like traits. Like *fka1-1* and *fka1-2, fka1-3* had dramatically reduced levels of *ARFTF17* transcripts (Supplementary Fig. 2k); an allelic test between *fka1-1* and *fka1-3* failed to complement the *fka1* phenotype in F₂ seeds (Fig. 2e and Supplementary Fig. 2l–n), confirming they are allelic.

## ARFTF17 regulates maize pericarp development

Since *ARFTF17* is highly expressed in the pericarp and the *fka1-1* phenotype is controlled maternally, we examined pericarp development in detail. The ovule integument is a single cell layer, while the fruit coat or pericarp develops from the ovary wall and typically contains multiple cell layers. Soon after double fertilization, the ovary wall begins to transform into pericarp tissue that rapidly increases in size by cell division and enlargement. By 12 DAP, pericarp cells can easily be characterized as two types: the outer cells have a flattened, ribbon shape with thick cell walls and are arrayed in layers, while the inner cells, with an amorphous, vesicular shape and thin walls, and are loosely organized with interlaced cell layers. The outermost layer is called the epidermis, the subtending cells the mesocarp, and the innermost cell layers the endocarp.

Light microscopy of semi-thin sections at 15 DAP and TEM observation of pericarp at 20 DAP revealed that the cell length, no matter on the crown or on the adgerminal sides of the *fka1-1* kernel, was significantly reduced compared to wild type (Fig. 3a–c and Supplementary Fig. 4a–d).The number of cells in *fka1-1* pericarp was also decreased (Fig.3d). In addition, the morphology of cells in *fka1-1* pericarp, particularly in the epidermis and mesocarp, was strikingly altered (Supplementary Fig. 4a, c). The shape of these cells became irregular, and their arrangement disordered. It appeared the *fka1-1* mutant lost the ability to maintain pericarp cell identity. At 20 DAP, TEM of pericarp cells at the crown of B73 kernels had organelles and

plentiful cytoplasm; in contrast, the cytoplasm in *fka1-1* pericarp cells became more condensed (Fig. 3e, f). At 30 DAP, little cytoplasm was found in *fka1-1* pericarp cells and all the organelles were degraded; however, organelles and cytoplasm were still plentiful in wild-type cells (Supplementary Fig. 4e, f). These observations suggest *fka1-1* pericarp undergoes programmed cell death (PCD) earlier than B73 pericarp. Toluidine blue staining showed the crown region of *fka1-1* pericarp was remarkably more permeable than that of B73 (Supplementary Fig. 4g, h), and the rigidity of *fka1-1* pericarp was enhanced (Supplementary Fig. 4i). Collectively, these results suggest mutation of *ARFTF17* affects multiple cellular features of the pericarp, including cell division, cell elongation, cell death, and cell wall rigidity.

## Changes in flavonoid synthesis and IAA content in *fka1-1* pericarp

ARFs are plant-specific transcription factors that function in auxin signaling to couple hormone perception with gene expression[18]. To investigate how *ARFTF17* regulates pericarp development, we performed transcriptome deep sequencing (RNA-seq) of B73 and *fka1-1* pericarps at 12, 20 and 30 DAP. After counting reads and their normalization, differentially expressed genes (DEGs) were identified (*P*-value < 0.05, absolute fold change > 1.5). We detected 4173, 3832, and 2978 DEGs in 12-, 20- and 30-DAP pericarps (Supplementary Data 1), respectively, of which 529 were identified as common DEGs (Supplementary Data 2; Fig. 3g). Kyoto Encyclopedia of Genes and Genomes (KEGG) analysis revealed DEGs at the three time points were highly enriched for phenylpropanoid biosynthesis and its related pathways, i.e., phenylalanine metabolism and flavonoid biosynthesis (Fig. 3h). Among the common DEGs, an auxin efflux transport PIN-FORMED gene, *PIN1* (*Zm00001d044812*), was obviously down-regulated in *fka1-1* pericarp at each development stage showing decreased FPKM values (Fig. 3i). Further studies showed that nine genes involved in the general phenylpropanoid pathway did not vary much between B73 and *fka1-1* pericarps at all developmental stages (Supplementary Data 3; Fig. 3i). In contrast, the expression of key genes in flavonoid synthesis, such as *CHS* (encoding chalcone synthase in the first step of flavonoid synthesis), *FLS* (encoding a flavonol synthase catalyzing formation of flavonols: kaempferol, quercetin and myrecetin), *DFR* (encoding a dihydroflavonol 4-reductasethat that catalyzes formation of phlobaphenemonomers) and four *UGTs* (encoding UDP-glycosyltransferases that catalyze glycosylation of flavonoids), were greatly enhanced in *fka1-1* pericarps by 30 DAP (Supplementary Data 3; Fig. 3i). The expression of these genes was confirmed by RT-qPCR using samples from kernels grown under different ecological conditions (Supplementary Data 4; Fig. 3i)

ZmPIN1-mediated polar auxin transport and IAA content play fundamental roles in maize seed development[19]. Down-regulation of *PIN1* could affect auxin content. Indeed, the IAA content in *fka1-1* pericarp was reduced at different growth locations (Fig. 3j). The transcriptome patterns led us to examine differences between the metabolomes of B73 and *fka1-1* pericarps. At 15- and 25-DAP, we identified 216 and 330 differential metabolites, respectively, in B73 and *fka1-1* pericarps (*P*-value < 0.05, absolute fold change > 2; Supplementary Data 5). Among the 159 common differential metabolites, 133 were flavonoids, and in *fka1-1* pericarps most flavonoids increased from several- to more than a thousand-fold between 15- and 25-DAP (Fig. 3k, l; Supplementary Data 6). The phenylpropanoid biosynthetic pathway and tryptophan-dependent IAA biosynthesis, summarized in Fig. 3m, share the common substrate chorismate[20–22]. The enhanced phenylpropanoid biosynthetic pathway might consume more chorismate, consistent with the reduced amount of indole, tryptophan, and IAA (Fig. 3m, j).

## ARFTF17 interacts with MYB40 to regulate down-stream genes

Transient expression of ARFTF17-eGFP in maize leaf protoplasts showed its localization in nuclei (Supplementary Fig. 5a), consistent

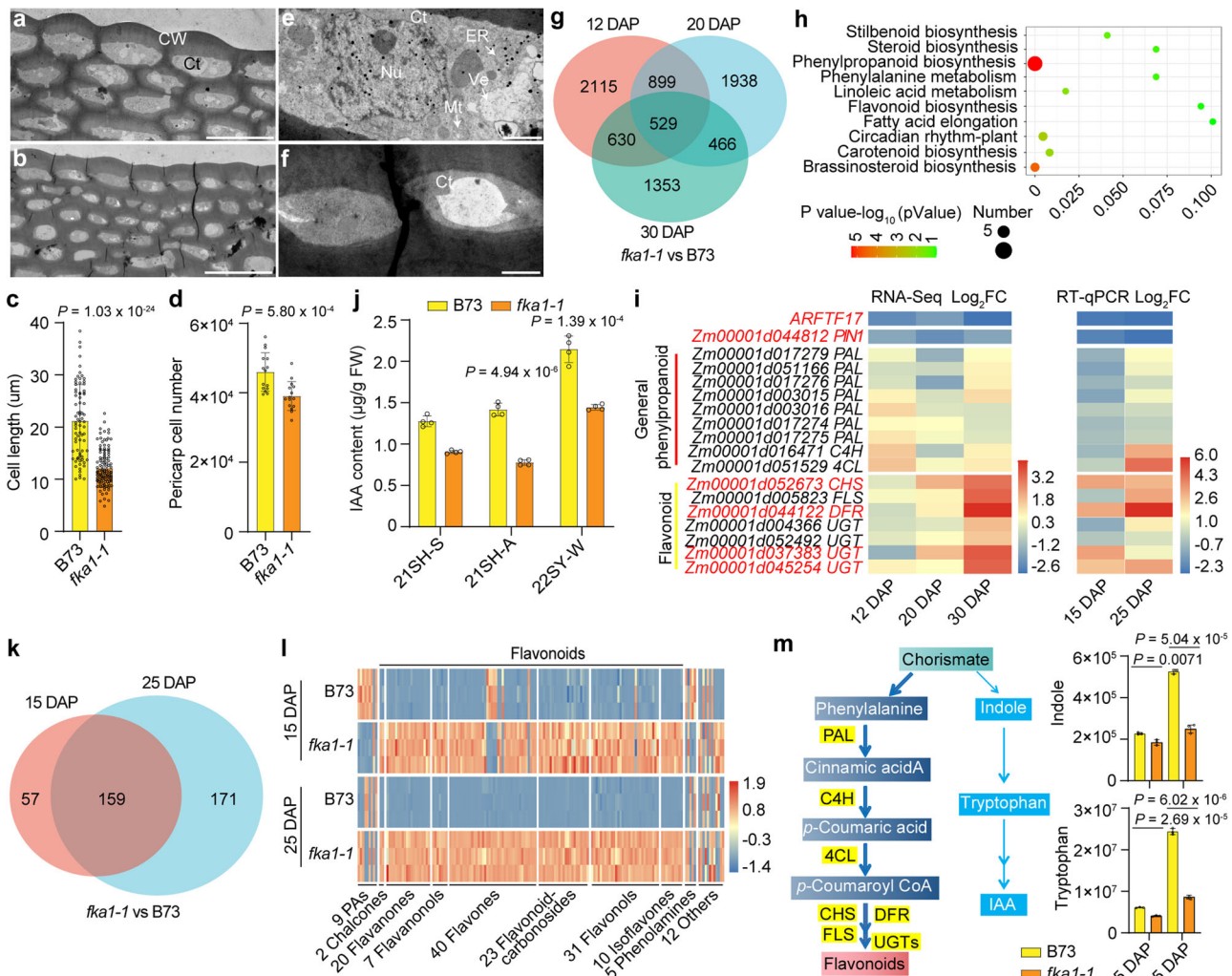

**Fig. 3 | Cellular, RNA-seq and metabolite analyses of B73 and *fka1-1* pericarps.**
**a**, **b** TEM observation of pericarp cells in kernel crown regions of B73 (**a**) and *fka1-1* (**b**) at 20 DAP. Scale bars, 20 μm. **c** Length of pericarp cells in kernel crown regions of B73 and *fka1-1* at 20 DAP. Data are mean ± s.d. (*n* = 76 cells in B73 and 116 cells in *fka1-1*). **d** Cell number in B73 and *fka1-1* pericarps at 15 DAP. Data are mean ± s.d. (*n* = 15 kernels). **e**, **f** Differences in cytoplasm of pericarp cells in the kernel crown regions of B73 (**e**) and *fka1-1* (**f**). Scale bars, 20 μm. **g** Venn diagram showing the intersection of DEGs determined by RNA-seq of B73 and *fka1-1* pericarps at 12, 20 and 30 DAP (*n* = 3 biologically independent samples). **h** KEGG enrichment analysis of DEGs in 12-, 20- and 30-DAP pericarps of B73 and *fka1-1*. **i** Expression of *PIN1* and genes involved in the phenylpropanoid pathway in B73 and *fka1-1* pericarps at 12, 20 and 30 DAP. Left panel, fold changes of differentially expressed genes from RNA-seq data. Three biologically independent samples were used; right panel, RT-qPCR

confirmation of the RNA-seq data. Expression level in B73 was set to one. Fold changes of *fka1-1* vs B73 were evaluated. The data were from three biological repeats. **j** IAA content in pericarps of B73 and *fka1-1* grown in 2021 Shanghai in Spring (21SH-S), 2021 Shanghai in Autumn (21SH-A) and 2022 Sanya in Winter (22SY-W). Data are mean ± s.d. (*n* = 4 biologically independent samples). **P < 0.01 (two-tailed Student's *t* test). **k** Venn diagram showing the intersection of differentially accumulated metabolites in B73 and *fka1-1* pericarps at 15 and 25 DAP. (**l**) Differences in accumulation of metabolites in B73 and *fka1-1* pericarps at 15 and 25 DAP. Three biologically independent samples were used. **m** A auxin and flavonoid biosynthesis share a common substrate. Relative contents of indole and tryptophan in metabonomics data of B73 and *fka1-1* pericarps. Data are mean ± s.d. (*n* = 3 biologically independent samples). Two-tailed Student's *t*-tests were used to determine *P*-values shown in the (**c, d, j, m**).

with ARFTF17 being a transcription factor. We performed an electrophoretic mobility shift assay (EMSA) and found that ARFTF17 did not bind to the *DR5* promoter or the P3(2x) fragment, the latter containing two tandem inverted repeats of the auxin response element (TGTCTC)[23] (Supplementary Fig. 5b, c). DNA Affinity Purification sequencing (DAP-seq) showed 14 maize ARFs preferentially bind to the core TGTC motif[17]; however the C clade members: ARFTF5, ARFTF19 and ARFTF21 had no unique peaks[17]. Based on this, we suspected ARFTF17 regulates gene expression through protein-protein interaction with an unknown transcription factor that can specifically recognize the downstream gene promoters.

Our data showed that ARFTF17 modulated flavonoid biosynthesis (Fig. 3i−m), and this pathway is known to be regulated in maize by *P* genes, which contain five members: *P1* (*Zm00001d028850*), *P2*

(*Zm00001d028842*), *MYB40* (*Zm00001d040621*), *MYB95* (*Zm00001d009088*) and *MYB154* (*Zm00001d047671*)[24–27]. P1 recognizes a CxxC core motif and transactivates the structural genes *c2* (*CHS*, encoding the chalcone synthase) and *a1* (*DFR*, encoding the dihydroflavonol 4-reductase)[25,26,28]. Based on our RNA-seq data, *P1*,*P2* and *MYB154* were not expressed in B73 pericarp. *MYB40* was intact and expressed at a high level in the pericarp, while the N-terminus of the MYB95 protein was missing and its transcript level greatly lower than that of *MYB40* (Supplementary Fig. 5d, e). Therefore, we speculated that ARFTF17 regulates flavonoid biosynthesis in B73 pericarp by interacting with MYB40. We performed bimolecular fluorescence complementation (BiFC), luciferase complementation imaging (LCI), and pull-down assays, and the results revealed an interaction between ARFTF17 and MYB40 (Fig. 4a−c). We then performed dual-luciferase

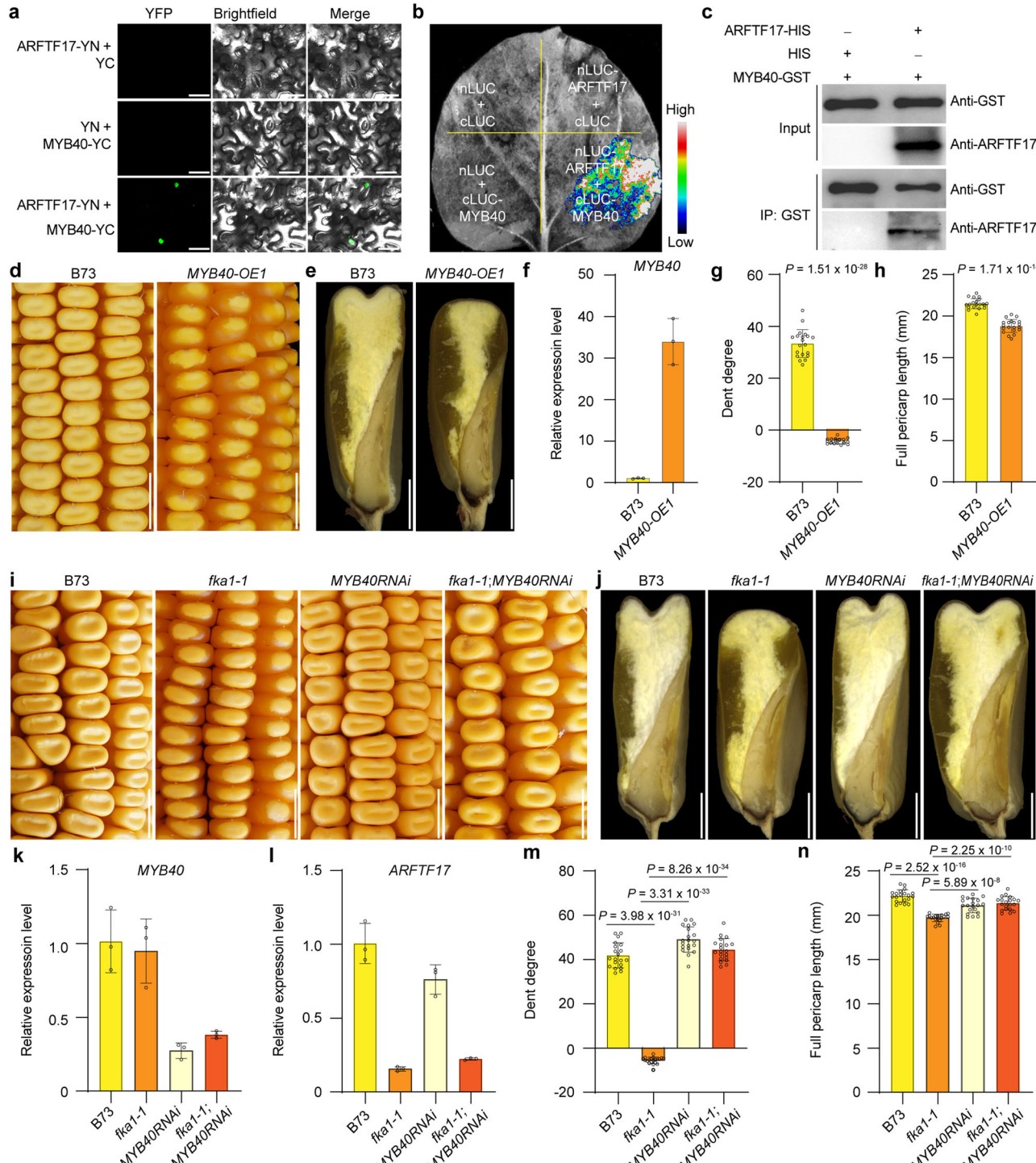

**Fig. 4 | ARFTF17 interacts with MYB40 to regulate pericarp development.**
**a** BiFC assay showing interaction between ARFTF17 and MYB40 in tobacco leaf protoplasts. Scale bars, 20 μm. **b** LIC assay showing interaction between ARFTF17 and MYB40 in a tobacco leaf. **c** In vitro pull-down assay showing interaction between ARFTF17 and MYB40. **d** The kernel phenotype of B73 and a *MYB40* over-expression line (*MYB40-OE1*). Scale bar, 1 cm. (**e**) Longitudinal-sections of mature kernels of B73 and *MYB40-OE1*. Scale bar, 2 mm. **f** RT-qPCR analysis of *MYB40* in pericarps of B73 and *MYB40-OE1*. **g, h** Dent degree of the kernel crown (**g**) and pericarp length (**h**) of B73 and *MYB40-OE1* at maturity. Data are mean ± s.d. (*n* = 20 kernels from 6 ears). **i** The kernel phenotype of B73, *fka1-1*, *MYB40RNAi* and *fka1-*

*1;MYB40RNAi*. Scale bar, 1 cm. **j** Longitudinal-sections of mature kernels of B73, *fka1-1*, *MYB40RNAi* and *fka1-1;MYB40RNAi*. Scale bar, 2 mm. **k, l** RT-qPCR analysis of *MYB40* (**k**) and *ARFTF17* (**l**) in pericarps of B73, *fka1-1*, *MYB40RNAi* and *fka1-1;MYB40RNAi*. **m, n** Dent degree (**m**) and pericarp length (**n**) of kernels of B73, *fka1-1*, *MYB40RNAi* and *fka1-1;MYB40RNAi* at maturity. Data are mean ± s.d. (*n* = 20 kernels from 6 ears). In (**f, k, l**), expression levels were normalized to that of *TUA4* (*Zm00001d013367*). Data are mean ± s.d. (*n* = 3 biologically independent samples). Two-tailed Student's *t* tests were used to determine *P* values shown in the (**j, h, m, n**).

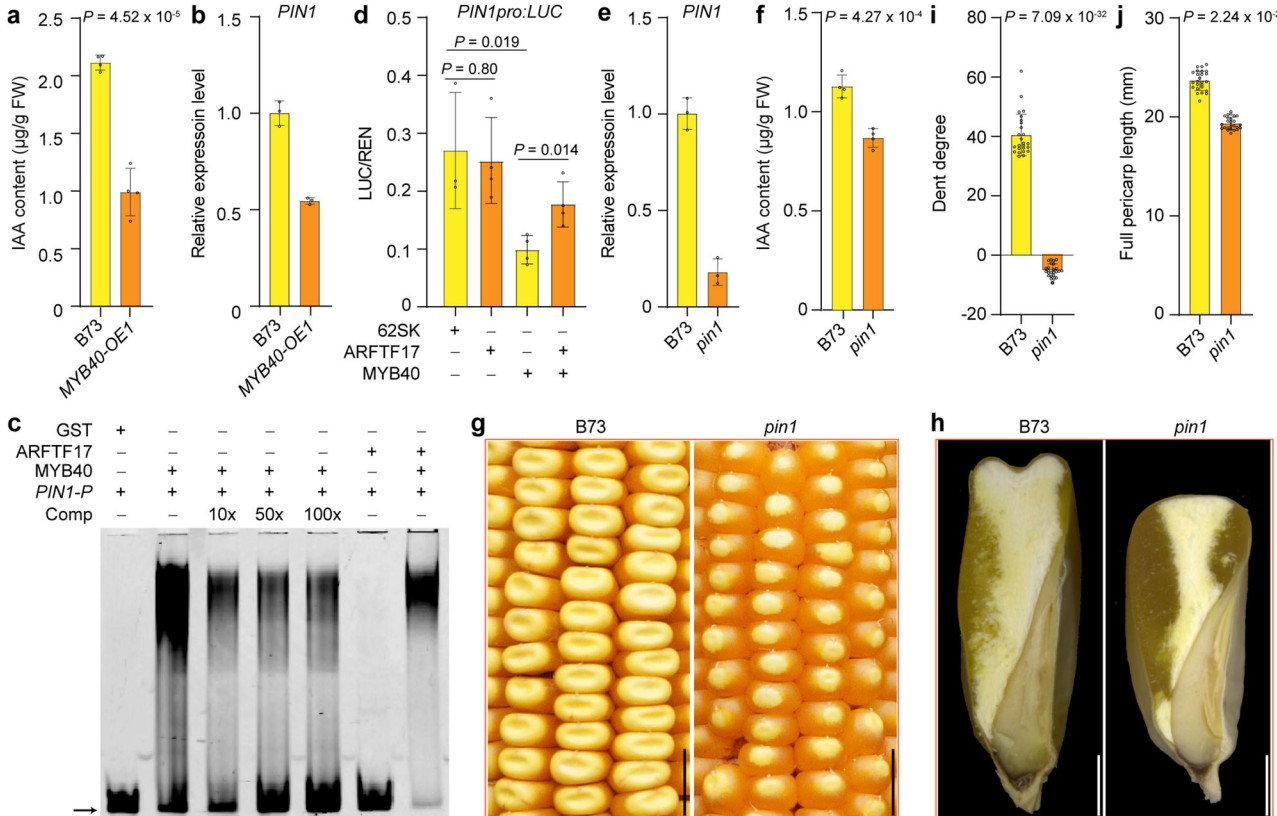

**Fig. 5 | MYB40 regulatesPIN1 and *pin1* shows flint like kernels. a** IAA content in pericarps of B73 and *MYB40-OE1*. Data are mean ± s.d. (*n* = 4 biologically independent samples). **b** RT-qPCR analysis of *PIN1* in pericarps of B73 and *MYB40-OE1*. Data are mean ± s.d. (*n* = 3 biologically independent samples). **c** EMSA showing MYB40 binds to the *PIN1* promoters. *PIN1-P*, the probe containing P1 binding cores positioned between −1015 and −883 bp upstream of the start codon in the *PIN1* promoter. Binding specificity was confirmed by competition, whereby the signals gradually decreased with addition of unlabeled wild-type probes (10, 50 and 100 x). **d** MYB40 repressing the *PIN1* promoter in maize leaf protoplasts. Data are mean ± s.d. (more than three biologically independent samples were performed).

**e** RT-qPCR analysis of *PIN1* expression in pericarps of B73 and *pin1*. Data are mean ± s.d. (*n* = 3 biologically independent samples). **f** IAA content in pericarp tissues of B73 and *pin1*. Data are mean ± s.d. (*n* = 4 biologically independent samples). **g** The kernel phenotype of B73 and *pin1*. Scale bars, 1 cm. **h** Longitudinal-sections of B73 and *pin1* kernels. Scale bars, 2 mm. **i** Dent degree of B73 and *pin1* kernels. Data are mean ± s.d. (*n* = 25 kernels from 6 ears). **j** Pericarp length of B73 and *pin1* kernels. Data are mean ± s.d. (*n* = 25 kernels from 6 ears). In (**b**, **e**), expression levels were normalized to that of *TUA4*. Data are mean ± s.d. (*n* = 3 biologically independent samples). Two-tailed Student's *t*-tests were used to determine *P*-values shown in the (**a**, **d**, **f**, **i**, **j**).

(LUC) assays to test whether MYB40 activates and ARFTF17 suppresses expression of *CHS* and *DFR*. We found co-expression of *35Spro:MYB40* with *CHSpro:LUC* or *DFRpro:LUC* in maize leaf protoplasts resulted in significantly increased LUC activity compared with the negative control (Supplementary Fig. 5f, g). When *35Spro:ARFTF17* was co-expressed with *35Spro:MYB40* and the reporters, the expression of LUC activities was markedly repressed (Supplementary Fig. 5f, g). EMSA revealed that MYB40 but not ARFTF17 bound *CHS* and *DFR* promoters that contained the P1 binding core motifs (Supplementary Fig. 5h, i). These results indicate that ARFTF17 regulates the downstream genes by interacting with MYB40 and the interaction represses MYB40-mediated gene transactivation.

### The ARFTF17-MYB40 module regulates flint-like kernel architecture

To genetically validate function of the ARFTF17-MYB40 module in regulating kernel architecture, we overexpressed *MYB40* in B73 using a maize *ubiquitin* promoter and obtained five independent transgenic lines. Similar to *ARFTF17* mutations, overexpression of *MYB40* created uniformly flint-like kernels on all the transgenic ears. As a representative, *MYB40-OE1* kernels manifested a convex crown containing more vitreous endosperm and reduced pericarp length (Fig. 4d–h). To confirm that the *fka1* phenotypes depend on the presence of *MYB40*, we created *MYB40-RNAi* lines in the B73 background, and then

introgressed *MYB40-RNAi* into *fka1-1*. We found that silenced expression of *MYB40* in the pericarp of the double mutant *fka1-1;MYB40RNAi* suppressed the flint-like kernel phenotype created by the *ARFTF17* mutation in the dent. *MYB40-RNAi* and *fka1-1;MYB40-RNAi* kernels showed similar dent degree and pericarp length to the wildtype B73 (Fig. 4i–n). These results support the hypothesis that *MYB40* is required for *ARFTF17* mutation-mediated flint-like kernel formation.

We measured the IAA content in *MYB40-OE1* pericarp and found that it was significantly decreased (Fig. 5a). Consistently, *PIN1* expression was down-regulated in *MYB40-OE1* pericarp (Fig. 5b), indicating MYB40 directly regulates *PIN1* expression. We found the *PIN1* promoter (−1015 to −883 bp upstream of the start codon) contained the P1 DNA binding cores. EMSA revealed that MYB40 but not ARFTF17 recognized the *PIN1* promoter (Fig. 5c). In contrast to *CHS* and *DFR* promoters, LUC assays showed that MYB40 repressed *PIN1* promoter activity (Fig. 5d). However, the function of MYB40-mediated repression on the *PIN1* promoter could be inhibited by ARFTF17, leading to partially enhanced LUC activity when ARFTF17, MYB40 and *PIN1Pro:-LUC* were co-expressed (Fig. 5d). These results indicate that MYB40 has dual functions of transactivation and repression, depending on the gene recognized. We identified a *pin1* EMS mutant in the B73 background from the MEMD database, in which the *pin1* mutation resulted from a stop-gain mutation at amino acid 467 in the PIN1 protein. The expression level of *PIN1* was significantly reduced in *pin1* pericarp, and

the IAA content was also decreased (Fig. 5e, f). *pin1* showed a flint-like kernel phenotype with a convex crown, more vitreous endosperm and a shorter pericarp length compared with wild type (Fig. 5g–j).

Based on the above results, we hypothesize that the ARFTF17-MYB40 module regulates pericarp length by regulating *PIN1* and flavonoid biosynthesis-related genes (*CHS* and *DFR*). Mutation of *ARFTF17* results in enhanced functions of MYB40, and as a consequence inhibits the expression of *PIN1*, prompting the expression of *CHS* and *DFR*. Besides reduced *PIN1* expression directly affecting IAA content, increased flavonoid biosynthesis could probably reduce the substrate available for IAA biosynthesis. These effects together caused decreased IAA content in *fka1* pericarps, which might in turn affect multiple cellular features of the pericarp and thus lead to a flint-like kernel phenotype (Supplementary Fig. 6). However, further experiments are needed to comprehensively unravel the underlying mechanism related to the role of auxin in the dent-flint kernel phenotype transition.

### *ARFTF17* mutation improves kernel architecture in dent inbreds and hybrids

To evaluate penetrance of the *fka1* phenotype in other dent backgrounds, we crossed *fka1-1* with 20 dent inbreds and observed the ear phenotypes of $F_2$ plants homozygous for *fka1-1*. Among them, 18 lines had a complete *fka1* phenotype, while only two lines (PHR55 and LH93) exhibited a partial *fka1* phenotype (Supplementary Fig. 7a). This confirmed the broad value for using *fka1* to improve dent kernel architecture. Subsequently, we backcrossed *fka1-1* into Zheng58 and Chang7-2 for four generations (Supplementary Fig. 7b). These two elite inbreds are used to make the Zhengdan958 hybrid, which has the largest corn growing area every year in the central region of China. When grown in Sanya, Zhengdan958-*fka1-1* (Fig. 6a) manifested a flint-like kernel phenotype, with increased vitreous endosperm (Fig. 6b–e). The kernels of Zhengdan958-*fka1-1* were more resistant to breakage (Fig. 6f). Importantly, the moisture content, a critical trait for grain harvesting and storage, of Zhengdan958-*fka1-1* kernels was significantly lower than that of Zhengdan958 (Fig. 6g). The test weight of Zhengdan958-*fka1-1* was also increased (Fig. 6h). Zhengdan958-*fka1-1* had essentially identical agronomical traits as Zhengdan958, including 100-kernel weight (Fig. 6i), kernel weight per ear (Fig. 6j) and the grain yield (Fig. 6k). We repeated the field experiment in Shanghai and found similar results (Supplementary Fig. 8). The results suggest introgression of *fka1-1* into many other dent hybrids has potential to improve their kernel quality, while lowering their moisture content at maturity.

## Discussion

The maize seed is a caryopsis, consisting of the filial embryo and endosperm and the surrounding seed coat, a maternal (sporophytic) tissue. Nutrients (primarily sucrose and amino acids) are drawn into pedicel cells at the base of the kernel and enter the endosperm through the basal transfer layer[29]. These nutrients move symplastically through endosperm cells, first into transfer cells, and eventually reaching large parenchyma storage cells, the so-called "starchy endosperm". There, most of the sucrose is converted into starch granules in amyloplasts, and the amino acids are used to synthesize storage proteins, which form accretions (protein bodies) within the lumen of the rough endoplasmic reticulum[30]. Midway through endosperm development (15-20 DAP), starchy endosperm cells in the center and crown regions of the kernel begin to undergo PCD[31]; eventually PCD affects all endosperm cells with the exception of the peripheral aleurone cell layer, which remains viable and produces hydrolytic enzymes during seed germination. Coincident with PCD, the contents of those starchy endosperm cells containing numerous protein bodies and starch granules begins to congeal, creating hard, translucent, vitreous endosperm, while those containing predominantly starch granules, form softer, opaque, starchy endosperm. The chemical process that creates vitreous endosperm is not well understood, but interactions between starch granules, protein bodies and the desiccating cell's cytoplasmic contents are involved, as air spaces are found between starch granules in the soft, starchy endosperm[32].

The fundamental differences between flint and dent kernel phenotypes are the shape and the degree to which vitreous endosperm is formed. Flint kernels are shorter and rounder, with a peripheral shell of vitreous endosperm that covers the crown and extends around the starchy central endosperm. Dent kernels tend to be longer, with the starchy center and crown flanked by vitreous regions. However, the spatial location and degree to which flints and dents form vitreous endosperm remain unknown.

Although little is known about the influence of the pericarp on cereal seed development, our studies demonstrate it has a surprising influence on several aspects of grain filling and desiccation in maize. It clearly plays an important role in development of the dent kernel phenotype. Presumably, pericarp enlargement is coordinated with endosperm growth. In flint types, expansion of the pericarp is more restricted, leading to shorter, fatter kernels, while in dents, its expansion outpaces growth of the endosperm. The more elongated phenotype of dent kernels might favor development of a starchy crown that collapses upon kernel desiccation (see below).

In view of the kernel phenotypes of *fka1-1* and B73, the pericarp does not appear to affect storage metabolite accumulation. Rather, it appears to influence formation of a denser, more vitreous kernel. This could be a consequence of the cellular structure of the pericarp, its earlier senesce, and perhaps its water permeability. The differences in pericarp structure of Zhengdan958-*fka1-1* and Zhengdan958 do not affect the kernel's sink strength or the metabolic activity of the endosperm and embryo, nor are there effects on phenotypic features of the ear, i.e., its length, kernel row number per ear, kernel number per row, and kernel weight per ear (Fig. 6). The degree to which the pericarp limits expansion of the seed coat and affects water loss during kernel desiccation, if indeed it does either or both, is unknown, but clearly, this merits investigation.

What is responsible for the crown structure of dent genotypes? We envision at least two possible explanations: 1) The longer pericarp of dent kernels extends beyond the top of the endosperm, so as the kernel begins to desiccate, it collapses, which contributes to the dent; 2) Apical cells of the endosperm contain fewer starch grains and protein bodies and other cytoplasmic contents. This could be a consequence of diminished symplastic movement of nutrients from the base to the crown of the kernel (Fig. 1f). In natural populations, the formation of dent and flint kernels may be far more complicated than can be explained solely from the perspective of pericarp development and grain filling. It is possible that hormone gradients, differential expression of metabolic genes, or physical properties of kernels may influence the formation of dent and flint kernels.

Auxin plays important roles in maize plant and kernel development. *vanishing tassel 2* (also known as *L-tryptophan--pyruvate aminotransferase 1*) has a defect in auxin synthesis and shows severe abnormalities in vegetative and reproductive development[33]. *sparse inflorescence 1* (a mutant of the *Yuc*-like gene encoding a flavin monooxygenase) displays auxin-deficient-related characteristics in the male inflorescence[34]. *defective endosperm 18* (a mutant of *ZmYuc1*) showed a reduced cell number, smaller endosperm cell volume, smaller seed size and reduced kernel weight[35,36]. In this study, we found that mutation of *ARFTF17* in B73 resulted in decreased IAA content during pericarp development. ARFs are plant-specific transcription factors functioning in auxin signaling and coupling hormone perception to gene expression. An ARF transcription factor generally contains an N-terminal B3-type DNA binding domain (DBD), an activation (AD) or repression domain (RD) in the middle region, and a carboxy-terminal domain (CTD) for dimerization or interaction with Aux/IAAs (auxin/indole-3-acetic acid family proteins)[18,37–39]. The DBD interacts with auxin response

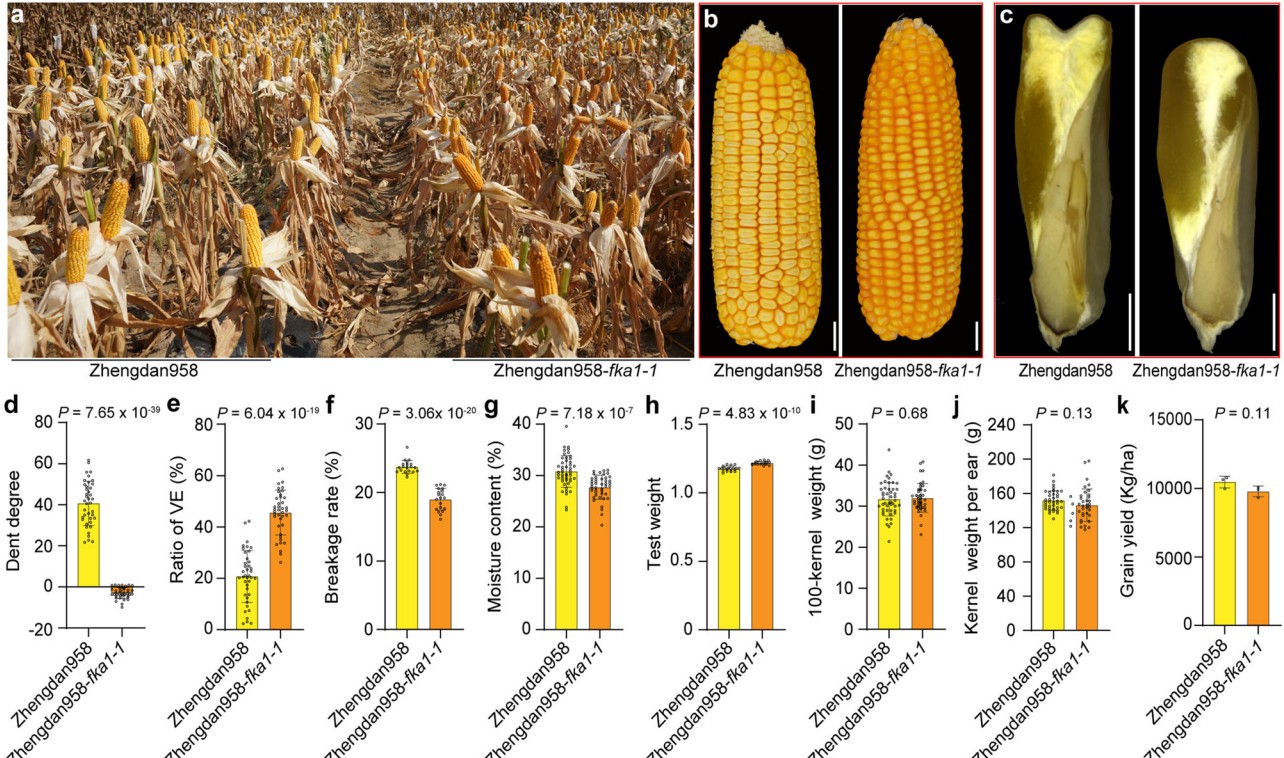

**Fig. 6 | Effect of *fka1-1* on the Zhengdan958 hybrid. a** Photograph of Zhengdan958 and Zhengdan958-*fka1-1* grown in field. **b** Ear phenotypes of Zhengdan958 and Zhengdan958-*fka1-1* hybrids. Scale bars, 1 cm. **c** Longitudinal-sections of Zhengdan958 and Zhengdan958-*fka1-1* kernels. Scale bars, 2 mm. **d** Measurement of kernel crown dent degree in Zhengdan958 and Zhengdan958-*fka1-1*. Data are mean ± s.d. ($n$ = 40 kernels from 20 ears). **e** Ratio of vitreous endosperm (VE) area to the total endosperm. Data are mean ± s.d. ($n$ = 40 kernels from 20 ears). (**f**) Breakage rate of kernels. Data are mean ± s.d. ($n$ = 20 ears). **g** Kernel moisture content of mature Zhengdan958 and Zhengdan958-*fka1-1*. Data are mean ± s.d. ($n$ = 48 Zhengdan958 ears and 44 Zhengdan958-*fka1-1* ears). **h** Test weight of Zhengdan958 and Zhengdan958-*fka1-1*. Data are mean ± s.d. ($n$ = 20 ears). **i** One hundred-kernel dry weight of Zhengdan958 and Zhengdan958-*fka1-1*. Data are mean ± s.d. ($n$ = 48 Zhengdan958 ears and 44 Zhengdan958-*fka1-1* ears). **j** Kernel weight per ear of Zhengdan958 and Zhengdan958-*fka1-1*. Data are mean ± s.d. ($n$ = 39 ears). **k** Grain yield of Zhengdan958 and Zhengdan958-*fka1-1*. Data are mean ± s.d. ($n$ = 3). Two-tailed Student's *t*-tests were used to determine *P*-values shown in the (**d**–**k**).

elements (AuxREs, TGTCTC/GAGACA)[23,38,40,41]. ARFTF17 has no AuxREs binding function, and its homologs in the C clade also have no specific DNA binding motif that is detectable in vitro[17], suggesting that additional factor(s) may be required for ARFTF17 DNA binding properties. We found that ARFTF17 can interact with MYB40 to regulate down stream genes. Whether an ARF transcription factor functions as an activator or a repressor appears to be determined by the amino acid sequence. AtARF5-8 and -19 are transcriptional activators[39]. They contain an AD enriched in glutamine (Q), serine (S), and leucine (L); by contrast, the AtARF2-4 and -9 repressors often contain a RD enriched in serine (S), proline (P) and leucine (L)[38,42]. The middle region of ARFTF17 lacks the Q-rich region, reminiscent of an ARF repressor[43].

MYB40 directly binds to the promoters of downstream genes, such as *CHS*, *DFR* and *PIN1*, and has dual functions of transactivation and repression. MYB40 activates *CHS* and *DFR* expression to promote flavonoid biosynthesis, but it represses *PIN1* expression. Because ARFTF17 inhibits MYB40, loss of function of *ARFTF17* or over-expression of MYB40 results in up-regulation of *CHS* and *DFR*, and down-regulation of *PIN1* in the pericarp, the latter directly reducing IAA content. Enhanced flavonoid biosynthesis may reduce metabolic flux in auxin biosynthesis, which can also reduce the IAA content in the pericarp. In addition, some flavonoids, such as apigenin, galangin, genistein, naringenin and rutinall, are known to be inhibitors of auxin transport[44–46]. Ultimately, IAA content is decreased during pericarp development in *fka1-1*, thereby leading to formation of seeds with a shorter pericarp than the *ARFTF17* wild type (Fig. 1g and Supplementary Fig. 1p). A more detailed explanation of these mechanisms will require further investigation.

Considering the degree to which the *arftf17* mutation phenocopies flint kernel traits in B73, we were surprised to find this gene was not associated with this phenotype in flint germplasm. It is possible *arftf17* creates flint-like kernel features in dent kernels by a unique mechanism, one unlike that found in flints. Alternatively, as a pleotropic mutation that affects multiple aspects of pericarp physiology, *arftf17* could be epistatic to the metabolic pathways influencing the small QTLs in native flint germplasm, thereby creating single gene control of this phenotype. While this explanation would be surprising, resolution of this question will require more extensive research than what we have done to date. Regardless, *arftf17* is a valuable tool with which to investigate the influence of the pericarp on maize kernel development and the utility of using *arftf17* to investigate the value of flint-like kernel features in elite dent hybrids.

Because of the diverse uses of maize as food and feed, an ideal kernel architecture would be one that provides the highest yield potential, while preserving seed integrity during harvesting and handling. The latter requires sufficient vitreous endosperm covering the kernel. Effective desiccation during kernel maturation is also a valuable trait, as dent hybrids typically have a high moisture content at harvest. Features of *fka1* have potential to improve the kernel phenotype of dent hybrids. When we incorporated *fka1-1* into the high yielding Zhengdan958 hybrid, it created denser, convex kernels with a flinty crown, similar to conventional flint hybrids. Importantly, the yield of Zhengdan958-*fka1-1* was comparable to the wildtype hybrid, plus the grain had a lower moisture content. Its increased seed integrity and more effective kernel desiccation would be desirable traits for other dent hybrids. Consequently, this research demonstrates great

potential for using mutation of *ARFTF17* to improve kernel architecture.

This investigation provides insight into important kernel traits, but there are several questions that merit further study. To unravel the genetic basis of natural dent and flint kernel phenotypes, it is necessary to obtain the full-genome sequence of multiple dent and flint maize inbreds and create specifically designed genetic populations for QTL mapping/GWAS studies. ARFTF17 influences multiple aspects of pericarp development, including cell division, cell elongation, cell death, cell wall rigidity, and auxin content, some or all of which could affect kernel architecture. Exploring the influence of each of them, in particular the role of auxin, is important for understanding their role in pericarp development and kernel architecture. This research will increase our understanding cereal seed development, with potential benefit for improving grain quality.

## Methods

### Plant materials and growth conditions

About 3 mL of B73 pollen was added to a 50 mL tube containing ethyl methane sulfonate (EMS, Sigma M0880) reagent diluted in mineral oil (1:1000, vol/vol). The mixture was shaken vigorously to suspend the pollen and then incubated for 45 min. During this time, the mixture was routinely shaken vigorously every minute (or continually if possible) to keep the pollen suspended. The treated pollen was applied to silks of B73 ears. The resulting $M_1$ seeds were planted and the surviving plants self-pollinated, yielding more than 2000 $M_1$ ears. Ten $M_2$ seeds from each $M_1$ ear were planted and the plants that grew were self-pollinated, producing only about 15,000 $M_2$ ears. By observing $M_3$ seeds on $M_2$ ears, we identified an ear with a uniform flint like kernel phenotype that we designated *fka1-1*. The *fka1-2* allele and *pin1* was obtained from the public EMS mutant library MEMD[15] (http://elabcaas.cn/memd/public/index.html#/) under the accession number EMS4-0718ab and EMS4-0b3dd4 respectively.

The *ARFTF* mutants were created in our lab using CRISPR/Cas9 in the Hi-II hybrid (B x A) via *Agrobacterium*-mediated transformation[47]. To synthesize a gRNA construct for simultaneous editing of *ARFTF2*, *ARFTF17*, *ARFTF19* and *ARFTF21*, we identified a target site in the conserved sequences of the four genes through the website (http://www.e-crisp.org/E-CRISP/designcrispr.html). The guide sequence, GTGCTTGCCAAGGACGTGCA, was inserted between the maize U6 promoter and terminator by a synthetic sequence *ARFTF-CRISPR* (Supplementary Table 1). The specific single, double, triple and quadruple *ARFTF* mutations were introgressed into B73 for four generations by backcrossing. The CRISPR/Cas9 construct was removed by segregation during backcrossing; this was verified using a specific pair of primers (Supplementary Table 1). After backcrossing, homozygous ears of the mutants were generated by self-pollination for two generations. After each generation of backcrossing and self-pollination, PCR and sequencing were used to select progeny that contained the desired *ARFTF* mutations.

To create the *ARFTF17Pro:GFP* construct, the 2151-bp promoter of *ARFTF17* was amplified using primers *ARFTF17Pro-F* and *ARFTF17Pro-R* and cloned into pCAMBIA3300. The GFP-encoding sequence was amplified using primers *GFP-CDS-F* and *GFP-CDS-R*, and then inserted downstream of the *ARFTF17* promoter. To create the *ARFTF17-Pro:ARFTF17* construct, *ARFTF17* CDS was fused with the Flag tag at the N-terminus using primers *ARFTF17-Flag-F* and *ARFTF17-CDS-R*, and then ligated to the *ARFTF17* promoter. To create the *MYB40* overexpression construct (*MYB40-OE*), the *MYB40* CDS was fused with the Flag tag in the N-terminal using primers *MYB40-FlagF* and *MYB40-CDS-R*, and then cloned into pCAMBIA3300 driven by the *ubi1* promoter (*ubiquitin1, Zm00001d015327*). To create the *MYB40RNAi* construct, a sequence containing the sense, intron and antisense segments was synthesized (Supplementary Table 1) and then cloned into pCAMBIA3300 driven by the *ubi1* promoter. Constructs *ARFTF17Pro:GFP*,

*ARFTF17Pro:ARFTF17*, *MYB40-OE* and *MYB40RNAi* were transformed into B73 via *Agrobacterium*-mediated transformation respectively by Wimi Biotechnology (Jiangsu). All primers used are listed in Supplementary Table 1.

All maize genetic materials were grown in the Song Jiang experimental field (30.5°N, 121.1°E) in Shanghai, China, and a field in Sanya (18.2°N, 109.3°E), China. Wild-type *N. benthamiana* was grown in a growth chamber with soil at 22 °C and 70% humidity under 16 h light and 8 h dark photoperiod using Philips TLD 36 W/865 and 36 W/830 bulbs (90 µmol/m²/s).

### Bulked segregant analysis (BSA)

Since the dent and flint traits are determined by the maternal genotype, an $F_{2:3}$ population ($F_3$ seeds from $F_2$ ears) of *fka1-1* and B73 was created to clone the *fka1* gene. About 600 $F_2$ seeds were planted, and the resulting plants numbered. A piece of leaf from each one was sampled for DNA extraction. By observing the kernel phenotype of $F_3$ seeds on $F_2$ ears, the $F_2$ plants were divided into two types: dent and flint like. DNA samples of 70 individuals of the respective phenotypes were pooled for BSA analysis by OE Biotech Co, Ltd (Shanghai, China). We used two methods for linkage localization analysis: SNP-Index analysis and the Euclidean Distance (ED) method. The SNP-index is a method for labeling association analysis by calculating the genotype frequency differences between the mixed pools, mainly to find the significant differences in genotype frequencies between the mixed pools, and using Δ(SNP-index) for statistics. The progeny SNPs determined the reads of the source of the mutant parents according to the genotype of the parents, and then calculated the SNP-index. The green and blue dots are SNP-index value, and the red line is the fitted line by the SNP-index sliding window, the orange line is the 95% confidence line, the purple line is the 99% confidence line.

The calculation method of ED (Euclidean distance) value according to the previous ref. 48, and the frequency distance of each mutation type between different mixed pools calculated, and the distance differences were used to reflect the linkage intensity between the marker and the target region. The higher the ED value, the stronger the linkage intensity of this region. We took the ED value calculated by this method to the power of six to further reduce the noise, and then, used the locally weighted smoothing (loess) regression method to fit the sixth power of the ED value. The line plot of ED[6] values was fitted using the LOESS method.

### Material identification

The *fka1-1* mutation site was identified by PCR amplification using primers *fka1-1-F* and *fka1-1-R* and sequencing. The mutation site of *fka1-2* was identified by PCR amplification using primers *fka1-2-F* and *fka1-2-R* and sequencing. To identify the genome editing sites, the specific primers for each *ARFTF* genes were designed for PCR amplification (*ARFTF2/17/19/21-crF*, *ARFTF2/17/19/21-crR*) and sequencing (*ARFTF2/17/19/21-crS*). The sequences were analyzed through the website (http://skl.scau.edu.cn/dsdecode/). The *pin1* mutation was identified by PCR amplification using primers *pin1-F* and *pin1-R* and sequencing. *ARFTF17Pro:GFP* transgenic plants were confirmed by PCR amplification using the primers *ARFTF17Pro-F1* and *GFP-R1*. *ARFTF17-Pro:ARFTF17* transgenic plants were confirmed by PCR amplification using primers *ARFTF17Pro-F1* and *ARFTF17-R*. *MYB40-OE* transgenic plants were confirmed by PCR amplification using primers *ubi1-F* and *MYB40-R*, and identified by the RT-qPCR using *MYB40-QF* and *MYB40-QR*. The *MYB40RNAi* mutation was identified by PCR amplification using primers *ubi1-F* and *RNAi-R* and sequencing. All primers used are listed in Supplementary Table 1.

### Phenotypic analysis

B73 and *fka1-1* kernels at different developing time points (15, 20, 25, 30 and 35 DAP) and maturity grown in Sanya were used for phenotypic

analysis. B73 and *fka1-1* kernels at 12 DAP, 15 DAP, 20 DAP, 25 DAP, 30 DAP and maturity were used for phenotypic analysis. At each time point, kernels from the middle of each ear were used to measure kernel length and kernel width using the WSeen's Automatic seeds test system (SC-G, Hangzhou, China). Kernels from each ear were used for longitudinal free-hand sectioning and the full pericarp length measured by Image J. The degree of dent for the kernel crown was measured by the "Angle tool" button of Image J. Kernels were desiccated at 65 °C for 100-kernels dry weight analysis, moisture content. Vitreous endosperm area was measured by Image J.

### Kernel breakage rate analysis

Kernel breakage rate was tested by a digital ultrafine grinding instrument PX-MFC90D (WIGGENS). About 20 g dry kernels for each sample (*m*) were used for analysis by a mill speed of 1200 r min$^{-1}$ and a pulverization time of 30 s. After sifted through the 1 mm sieve, the mass on the sieve (*m1*) were recorded to calculate the kernel breakage rate by a formula: Breakage rate (%) = (*m-m1*)/*m* x100. The mature kernels of B73, *fka1-1*, Zhengdan958 and Zhengdan958-*fka1-1* grown in Shanghai and Sanya were used for analysis.

### Cytological analysis

Six ears of B73 and *fka1-1* at 15, 20, and 30 DAP were used for semi-thin sectioning. Longitudinal sections of 1 mm from developing kernels were fixed in FAA buffer (formaldehyde: acetic acid: ethanol: water = 10:5:50:35, v/v/v/v) and vacuumed infiltrated twice for 30 min. After dehydration using a gradient concentration of ethanol, the sections were embedded in epoxide resin for semi-thin sectioning. The sections were stained with 0.1% toluidine blue solution and then photographed under bright field using an ECLIPSE 80i microscope (Nikon, Tokyo, Japan). Cell length of the pericarp at 15 DAP was measured using Image J.

For TEM observation, about 2 mm longitudinal sections of the kernel with pericarp at 20 and 30 DAP were excised and immediately fixed in 2.5% glutaraldehyde in phosphate buffer, pH 7.2, then dehydrated and embedded following the standard protocol[49]. Ultrathin sections of the samples were cut with a diamond knife on a Leica EMUC6-FC6 ultramicrotome (USA) and imaged using Hitachi H-7650 transmission electron microscope (Japan).

### Tissue staining

To detect changes in the composition of B73 and *fka1-1* pericarp cell walls, we stained the seeds with toluidine blue[33]. The seeds were carefully dissected to avoid damaging the pericarp and were stained with 5 mL 0.1% toluidine blue in a 15 mL tube for two hours. After washing five times using distilled water, the pericarp covering the crown region was cut off using a double-edge blade. The pericarp section was placed in a 2 mL Eppendorf tube and homogenized with 0.5 mL of 80% alcohol. After extraction for two hours on a shaker, the homogenate was centrifuged at 17,370 g for five min. A sample of 0.5 mL of the supernatant was analyzed by measuring the absorbance at 626 nm. Unstained pericarp from the crown region was used as negative control. Different concentrations of toluidine blue were diluted, and a standard curve was created to quantify the content of toluidine blue in the samples.

### Flow cytometric analysis of the pericarp cell number

Flow cytometry was performed as previously described in ref. 50. Six ears each of B73 and *fka1-1* at 15 DAP were used for analysis. Using a tweezer, the entire pericarp was carefully peeled from each kernel without including any contaminating endosperm and embryo tissues. Three pericarps were pooled as one sample in a 5 cm plastic petri dish, and 1 mL of ice-cold Galbraith's extraction buffer (45 mM MgCl$_2$, 20 mM MOPS, 30 mM sodium citrate, 0.1% [v/v] Triton X-100, 0.05 mM sodium metabisulfite, 0.5% [v/v] -Mercaptoethanol, pH 7.0)[51], added, followed by rapid chopping with a new razor blade. The sample was filtered through a 40 mm nylon mesh into a 1.5 mL sample tube. The

filtered sample was added to DAPI buffer (5 μg/mL) and cells measured immediately using a flow cytometer CytoFLEX LX (Beckman Coulter). Data were collected and analyzed using CytExpert (Beckman Coulter) software. This analysis was repeated 18 times.

### RNA extraction and expression analysis

Total RNA was extracted using TRIzol reagent (Invitrogen, catalog number: 15,596,018) and purified with a RNeasy Mini Kit (Qiagen, catalog number: 74,106). After DNaseI digestion (Qiagen, catalog number: 79,254), RNA was used for reverse transcription by a SuperScript III First Strand Synthesis Kit (Invitrogen, catalog number 18,080,051). About 80–100 μg cDNA of each sample was used for RT-qPCR with SYBR Green (TAKARA) on a Bio Rad CFX-96 thermocycler.

To analyze the spatial expression pattern of *fka1-1*, B73 root, stem and leaf tissues at the flowering stage, and the pericarp and endosperm from developing kernels were collected for RNA extraction. The maize *SGT1* (*Zm00001d044172*) gene was used as an internal control to normalize the relative quantification of gene expression using the comparative CT method (ΔΔCt method).

For RNA-seq analysis, the pericarp tissues of B73 and *fka1-1* at different developmental time points were peeled with tweeter and immediately placed in liquid nitrogen, and then stored in a −80 freezer. After DNaseI digestion, RNA was used for the library construction, sequencing and data analysis by Shanghai OE Biotech Co, Ltd.

To analyze the expression level of *ARFTF17*, *PIN1*, *MYB40* and genes involved in phenylpropanoid pathway, the pericarp tissues were collected from the developing kernels of corresponding genetic materials. The maize *TUA4* (*Zm00001d013367*) gene was used as the internal control to normalize quantification of gene expression using the comparative CT method. All data were generated from three biological replicates of each sample. The primers used for RT-qPCR are listed in Supplementary Table 1.

### Antibody preparation and immunoblotting analysis

A partial ARFTF17 protein fragment from the 480th to 644th amino acid was used by ABclonal (Wuhan, China) to make antibodies. To analyze accumulation of ARFTF17 protein in B73 and different *fka1-1* mutants, total protein was extracted from the pericarp and endosperm at eight and 12 DAP with the non-zein buffer[52]. Twenty μg of total protein was separated by 10% SDS-PAGE and transferred electrophoretically to a PVDF membrane. Immunodetection used ARFTF17 antibodies at a concentration of 1:1,000 at 4 °C overnight, followed by a secondary anti-rabbit-HRP at a concentration of 1:5,000 (Abmart, catalog number: M21002L). To detect the control protein, ACTIN, a primary antibody, mouse monoclonal ACTIN antibody (Abmart, catalog number: M20009L) and a secondary antibody, anti-mouse IgG-HRP (Abmart, catalog number M21001L), were used. To examine FLAG in *fka1-1*; *ARFTF17pro:Flag-ARFTF17* plants, total protein was extracted from the pericarp. Immunoblotting used anti-FLAG (Sigma, A8592) as the primary antibody at a dilution of 1:1,000, and anti-mouse IgG-HRP (Abmart, M21001L) as the secondary antibody at a dilution of 1:5,000. The membranes were treated with a chemiluminescence substrate (Invitrogen, catalog number: WP20005), and the immunoreactive bands were detected using Tanon-5200 system.

### Subcellular localization

To determine the subcellular localization of ARFTF17, the full-length coding sequence without the stop codon was amplified using primers *ARFTF17-GFPF* and *ARFTF17-GFPR*, and cloned into the pCAMBIA1300-GFP vector to generate the *35Spro: ARFTF17-GFP* fusion plasmid. B73 seedlings were grown at 28 °C for seven d in the dark. Then the leaves were cut into one mm sections and incubated in a maceration buffer (1.5% Cellulase R10, 0.5% Macerozyme R10, 0.4 M Mannitol, 20 mM KCl, 20 mM MES pH 5.7, 10 mM CaCl$_2$, 0.1% BSA, 5 mM β-Mercaptoethanol). After 3 h at 22 °C, W5 buffer (154 mM NaCl, 125 mM

CaCl$_2$, 5 mM KCl, 5 mM Glucose, 0.03% MES, pH 5.7) was used to terminate the reaction and wash and collect the protoplasts. Then, MMG buffer (0.4 M Mannitol, 15 mM MgCl$_2$, 0.1% MES, pH 5.7) was added to adjust the concentration of protoplasts to $1 \times 10^6$/mL. For every 100 μL of protoplast buffer, 10 μg of plasmid was added and 110 μL of PEG (45% PEG4000, 0.2 M Mannitol, 100 mM CaCl$_2$) for transformation. After incubation at room temperature for 15 min, 440 μL of W5 buffer was added for termination. After removing the supernatant, 1 mL of WI (20 mM KCl, 0.6 M Mannitol, 4 mM MES pH5.7) was added and the protoplasts incubated for 16 h. The GFP fluorescent signal was observed with 488 nm excitation and 510 nm emission under an LSM880 confocal microscope (Zeiss, Jena, Germany).

### Metabolome analysis

The metabolome was analyzed by MetWare (http://www.metware.cn/) based on UPLC (Ultra Performance Liquid Chromatography UPLC)-MS/MS (Tandem mass spectrometry). The pericarp tissues of B73 and *fka1-1* at 15 and 20 DAP were used for analysis. After freeze-drying with a vacuum freeze-dryer (Scientz-100F), a 100 mg sample was extracted six times with 1.2 mL of 70% methanol solution, and then the sample was placed in a refrigerator at 4 °C overnight. Following centrifugation at 13,523 g for 10 min, the extracts were filtered (SCAA-104, 0.22 μm pore size; ANPEL, Shanghai) and analyzed with a UPLC system (UPLC, SHIMADZU Nexera X2) using an Agilent SB-C18 column (1.8 μm, 2.1 mm * 100 mm). The flow velocity was 0.35 mL per min at 40 °C; the injection volume was 4 μL. The effluent was alternatively connected to an ESI-triple quadrupole-linear ion trap (QTRAP)-MS (AB4500 Q TRAP UPLC/MS/MS System) equipped with an ESI Turbo Ion-Spray interface, operating in positive and negative ion mode and controlled by Analyst 1.6.3 software (AB Sciex).

### Measurement of auxin content

To measure the IAA content, pericarp tissues were collected and immediately placed in liquid nitrogen, and then stored in a −80 freezer. A 50 mg sample was extracted in 1 mL of 80% methanol solution for 4 h, and then centrifuged at 13,523 g at 4 °C for 10 min. The liquid supernatants were placed at −20 °C overnight and centrifuged at 13,523 g at 4 °C for 10 min before analysis. The extracts were analyzed on a UPLC instrument combined with a QTRAP® 6500 + MS system equipped with an electrospray ionization (ESI) source (AB SCIEX). Instrument control and data acquisition were performed using Analyst 1.6.3 software (AB SCIEX), and data processing was performed using MultiQuant 3.0.2 software (AB SCIEX). For quantifying the IAA content, 0, 5, 20, 50, 200, and 500 ng/mL IAA (Agrisera) were used to create a standard curve.

### Bimolecular fluorescence complementation assays (BiFC)

The full-length coding sequences of *ARFTF17* and *MYB40* were amplified and inserted into nYFP and cYFP plasmids to obtain ARFTF17-YN and MYB40-YC vectors, respectively. The vectors were first transformed into *A. tumefaciens strain* GV3101. The strains containing the indicated vectors were collected by centrifugation and resuspended in infiltration buffer (10 mM MgCl$_2$, 10 mM MES, and 150 mM acetosyringone) at final OD600 = 1.0, then mixed in a 1:1 ratio and injected into *N. benthamiana* leaf epidermal cells. The BiFC-induced YFP fluorescence was detected by confocal laser scanning microscopy (LSM880; Zeiss) after 48 h post inoculation. The primers used in BiFC assays are listed in Supplementary Table 1.

### Luciferase complementation imaging assay (LCI)

The coding sequences of *ARFTF17* and *MYB40* were cloned into JW771 (NLUC, N-terminal half) and JW772 (CLUC, carboxyl-terminal half)[53], respectively. The empty JW771 and JW772 vectors were used as the negative controls. *Agrobacterium tumefaciens* GV3101 cells were resuspended in infiltration buffer (10 mM of MgCl$_2$, 10 mM of MES, and 150 mM of acetosyringone) at an OD600 between 0.8 to 1. The suspension was infiltrated into the abaxial side of *N. benthamiana leaves*. After incubation for 2 days, luciferin was infiltrated into the same region of the leaves. Then, the Tanon-5200 Chemiluminescent Imaging System (Tanon Science and Technology) was used to monitor luciferase signals. The primers used in LCI assays are listed in Supplementary Table 1.

### Recombinant protein preparation and pull-down assay

The coding sequence of *ARFTF17* was amplified using primers *ARFTF17-PET30F* and *ARFTF17-PET30R* and cloned into the pET30a vector to create a fusion protein with a His tag. The coding sequence of *MYB40* was amplified using primers *MYB40-pCOLDF* and *MYB40-pCOLDR*, and cloned into the pCold vector to create a fusion protein with the GST tag at NH-terminus. Recombinant proteins were produced in *Escherichia coli* BL21 (DE3) (EC1002) induced with 0.5 mM isopropyl β-D-thiogalactopyranoside (IPTG) at 20 °C for 20 h. The proteins were purified on Ni-NTA agarose (QIAGEN, 30210), following the manufacturer's instructions. For pull-down assays, equal amounts of bait and prey proteins were incubated at 4 °C overnight in binding buffer (20 mM Tris-HCl, pH 8.0, 150 mM of NaCl, 0.2% [v/v] Triton X-100, 10% (v/v) glycerol, EDTA free protease inhibitor1025 cocktail (Roche, 4693132001). The protein complexes were recovered with glutathione Sepharose beads (Sangon Biotech, C650031), which were washed three times with buffer (50 mM Tris-HCl, pH 8.0, 140 mM of NaCl, 0.1% [v/v] Triton X-100, and EDTA free protease inhibitor1025 cocktail (Roche, 4693132001). After removing unbound proteins, those bound were analyzed by immunoblotting using anti-GST (Abmart, M20007) and anti-ARFTF17 antibodies. The primers used in this assay are listed in Supplementary Table 1.

### Dual-luciferase reporter assay (DLR) in maize protoplasts

The coding sequences of *ARFTF17* and *MYB40* were cloned into the effector construct, 62SK, driven by a 35 S promoter. The promoters of *CHS*, *DFR* and *PIN1* were cloned upstream of the *LUC* gene in the reporter vector pGreenII 0800. The plasmid concentration was adjusted to 1 μg/μL. Protoplast extractions and transformations were described above. After incubating 16 h, total protein was extracted from the transformed protoplasts and analyzed on a luminometer (Promega 20/20) using a commercial LUC analysis kit. At least three biological replicates were performed for each experiment. The primers used in DLR assays are listed in Supplementary Table 1.

### Electrophoretic mobility shift assay (EMSA)

The recombinant proteins of ARFTF17-His andMYB40-GST were purified as described above. The oligonucleotide probes (100 bp) containing the CxxC element from the target gene promoters (*CHS,DFR* and *PIN1*) were 5′ end-labeled with fluorescein amidite (FAM), which was synthesized by BioSune Biotech Co, Ltd. Labeled probes of 0.05 pmol were incubated with 100 ng of purified recombinant proteins at room temperature for 30 min in a binding reaction mixture of 20 μL (1 μL of 1 mg/mL salmon sperm DNA, 20 mM Tris-HCl, pH 7.9, 5% glycerol, 0.04 mg/mL bovine serum albumin, 2 mM MgCl$_2$, 0.2 mM dithiothreitol and 40 mM KCl, ultrapure water to final volume of 20 μL). For competition assays, 10-, 50- and 100-fold unlabeled probes were added to the reaction mixture. Electrophoresis was in a native 4% polyacrylamide gel for 80 min with 0.5 M Tris-borate-EDTA at 4 °C and constant 110 voltage. Fluorescence was detected with a Starion FLA-9000 instrument (FujiFilm, Japan). The primers used in EMSA assays are listed in Supplementary Table 1.

### Introgressing *fka1-1* into inbreds and Zhengdan958 hybrid

To evaluate the phenotypic effects of *fka1-1*, we introgressed *fka1-1* into 20 dent inbred lines and created F$_2$ populations. For each population, we planted the F$_2$ and identified the *fka1-1* genotype by PCR amplification and sequencing. Then we obtained homozygous *fka1-1*

**Article** https://doi.org/10.1038/s41467-024-46955-9

plants and the corresponding WT genotype and self-pollinated them for phenotype observation.

The *fka1-1* allele was introgressed into the parental inbred lines of Zhengdan958 (Zheng58 and Chang7-2) by backcrossing four generations. Homozygous Zheng58-*fka1-1* and Chang7-2-*fka1-1* seeds were selected from BC₄F₂ populations. In each generation, the *fka1-1* mutation was monitored by PCR amplification and sequencing. The hybrid was created by crossing Zheng58-*fka1-1* and Chang7-2-*fka1-1*.The kernel phenotypes were analyzed as above described.

For the measurement of grain yield, Zheng58-*fka1-1* and Chang7-2-*fka1-1* were planted in a six-row plot, and rows were 0.5 m apart, with a planting density of 63,000 plants/ha. 3 plots were performed for each material. All plants were open-pollinated. After maturity, the kernel dry weight of 40 ears were quantified in each plot.

### Statistics and analysis

GraphPad Prism 8 was used for statistics analysis for one-way ANOVA and Tukey's test. Microsoft Excel 2016 was used for two-side Student's *t*-test analysis.

### Reporting summary

Further information on research design is available in the Nature Portfolio Reporting Summary linked to this article.

## Data availability

All data are available in the main text or the supplementary materials. Data for each figure in this paper are provided in the Supplementary Data files and a Source Data file. Source data are provided with this paper.

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

## Acknowledgements

We thank Zhiping Zhang, Jiqin Li, Shanshan Wang, Lianyan Jing, Xiaoyan Xu, Lina Xu. and Wenli Hu (CAS Center for Excellence in Molecular Plant Sciences, CAS) for technical support. We thank Li Qin (Qilu Normal University) for material support. We thank Xing Zeng, Zhenhua Wang and Lin Zhang (Northeast Agricultural University) for help with maize planting in Harbin. This research was supported by the STI 2030-Major Project (2023ZD0406901 to HW and YW), the National Natural Science Foundation of China (31925030 to YW), and the Ministry of Science and Technology of China (2022YFF1003302 to YW and WW).

## Author contributions

Y.W., H.W., Y.H. and W.W. designed research. Y.W. supervised the project. H.W., Y.H., Y.C., Y.L., X.X., Y.Z., Q.W., X.W., G.M., X.L., Q.X., X.H. and J.W. contributed to genetic material constructions, genotype analysis and phenotype analysis. H.W., Y.H. and Y.L. contributed to gene cloning, metabonomic analysis, and IAA measurement. Y.L., Y.C. and H.W. contributed to protein interaction, EMSA, transcriptional activation. H.W., Y.C., Y.L. and X.G. contributed to cytological experiments and data analysis. Y.W., H.W., Y.H., W.W. and B.L. analyzed the data, drafted and edited the manuscript.

## Competing interests

The authors declare no competing interests.
