## [Peer Review File · Nature Communications]

REVIEWERS' COMMENTS

Reviewer #1 (Remarks to the Author):

This submission reports the identification of a mutation that converts the dent phenotype of reference line B73 to a flint phenotype. The mutation was found to act in the pericarp and upon molecular mapping identified as a transcription factor, namely an auxin response factor that is expressed highly in the pericarp. An additional allele was obtained that showed no complementation but transgenes expressing the candidate gene did complement the mutation. Gene editing produced an additional allele. Gene expression studies found that PIN1, an auxin transporter, was expressed at a lower level in the mutant but genes in the flavonoid pathway were increased in the pericarps. An interaction was found with another transcription factor, MYB40. Various experiments indicated that the ARFTF17 mutation would result in MYB40 inhibiting PIN1 expression and enhancing the expression of chalcone synthetase and DFR. The natural variation between dent and flint varieties does not appear to involve the studied single gene mutation. However, it was demonstrated that the single gene could effectively make this conversion, and this has advantages for maize cultivation.

I found the paper to be very clear with a logical progression of presentation. The conclusions are supported by strong data. The story is very interesting and has important translational value.

Some abbreviations such as DAP, PCD, and DBD do not appear to be defined on first usage. Most readers will likely know, but it would be good to define for the more general reader.

Line 88, needs a semicolon rather than a comma.

Line 221, the usual designation of the transformation line is "Hi-II".

Line 324, needs a semicolon rather than a comma.

Lines 494-499, this reviewer wonders if the word "suppressor" should be replaced by "repressor". The word suppressor in genetic terminology means to correct one mutation by another back to normal. I don't think this is the intended meaning.

Lines 520-523, it would be best of leave out projected experimental ideas.

This reviewer wonders why if CHS and DFR are upregulated in mutant pericarps, why these pericarps do not express pigment, especially since most flavonoids were increased. What does the RNAseq data reveal about other pigment pathway genes needed?

Some comments for the authors, not for this submission but for future work, follow. With regard to their metabolic flux hypothesis that an enhanced phenylpropanoid pathway shifts away from the production of auxin, would a dominant negative mutation of chalcone synthetase, namely c2-Idf, reverse the effect of the mutant? On the other hand, would the r1-cherry allele that produces extremely high levels of anthocyanin in the pericarp reverse the dent phenotype? Several accessions of r1-cherry in the reviewer's collection all show a flint phenotype but this does not necessarily mean anything. A test of whether r1-cherry can convert B73 to flint would be a valid test.

Reviewer #2 (Remarks to the Author):

What a fantastic manuscript! This study identifies a novel genetic determinant of flint kernel architecture and provides an exhaustive set of data, from molecular to genetic, establishing a role for an auxin response factor (ARFTF17) in maize kernel development. This manuscript was a joy to read and is sure to be highly-cited. The data was very high quality and very exciting.

Minor comments for suggested edits:

1. Line 36: Indicate which PIN1 gene(s) are regulated by ARFTF17 (there are several in maize).
2. Lines 99 & 105: Suggest re-wording for clarity regarding the nature of the ARFTF17 mutation (missense, nonsense, knock-down, null, etc). Are the phenotypes due to altered ARFTF17 function, structure, or expression level?
3. Data S4 – add non-ambiguous gene identifiers for ARFTF17 and PIN1 (there are several PIN1 genes in maize and ARF annotation is variable depending on which phylogeny and genome annotation version is being referenced).
4. Line 152-153 should be moved to the discussion.
5. Lines 468-469: could the authors speculate on that contributes to apical patterning in the kernel during development? For example, could this pattern be established due to a hormone gradient (such as auxin) or differential expression of metabolic genes or physical properties of the kernel? Does the integument play a role in setting up the proximal-distal pattern?
6. Line 491-492: Suggest changing the phrase “indicating ARFTF17 cannot directly regulate its downstream target genes” to “suggesting that additional factor(s) may be required for ARFTF17 DNA binding properties.”
7. Line 492: change “needs to interact” to “can interact”.
8. Line 550: change the word “study” to “investigation”.
9. Add scale bars to all micrographs in figures (i.e. Figure 1a,b,f; Fig 2c,e; Fig 3a,b,e,f; Fig 4a,d,e,i,j; Fig 5g,h; Fig 6b,c;) and extended figures.
10. Line 1130: Indicate amount of cDNA template used in each RT-qPCR.
11. Line 1183-184: Indicate the excitation and emission wavelengths used for imaging GFP.

REVIEWERS' COMMENTS

Reviewer #1 (Remarks to the Author):

This submission reports the identification of a mutation that converts the dent phenotype of reference line B73 to a flint phenotype. The mutation was found to act in the pericarp and upon molecular mapping identified as a transcription factor, namely an auxin response factor that is expressed highly in the pericarp. An additional allele was obtained that showed no complementation but transgenes expressing the candidate gene did complement the mutation. Gene editing produced an additional allele. Gene expression studies found that PIN1, an auxin transporter, was expressed at a lower level in the mutant but genes in the flavonoid pathway were increased in the pericarps. An interaction was found with another transcription factor, MYB40. Various experiments indicated that the ARFTF17 mutation would result in MYB40 inhibiting PIN1 expression and enhancing the expression of chalcone synthetase and DFR. The natural variation between dent and flint varieties does not appear to involve the studied single gene mutation. However, it was demonstrated that the single gene could effectively make this conversion, and this has advantages for maize cultivation.

I found the paper to be very clear with a logical progression of presentation. The conclusions are supported by strong data. The story is very interesting and has important translational value.

Response: Thanks for your comments.

Some abbreviations such as DAP, PCD, and DBD do not appear to be defined on first usage. Most readers will likely know, but it would be good to define for the more general reader.

Response: Thanks for your suggestion. These have been revised.

Line 88, needs a semicolon rather than a comma.

Response: Thanks for your suggestion. This has been revised.

Line 221, the usual designation of the transformation line is "Hi-II".

Response: Thanks for your suggestion. This has been revised.

Line 324, needs a semicolon rather than a comma.

Response: Thanks for your suggestion. This has been revised.

Lines 494-499, this reviewer wonders if the word “suppressor” should be replaced by “repressor”. The word suppressor in genetic terminology means to correct one mutation by another back to normal. I don’t think this is the intended meaning.

Response: Thanks for your suggestion. This has been revised.

Lines 520-523, it would be best of leave out projected experimental ideas.

Response: Thanks for your suggestion. This section has been deleted.

This reviewer wonders why if CHS and DFR are upregulated in mutant pericarps, why these pericarps do not express pigment, especially since most flavonoids were increased. What does the RNAseq data reveal about other pigment pathway genes needed?

Response: Thanks for your questions.

Some maize inbreds accumulate red phlobaphene pigments in the pericarp. Phlobaphene is polymerized by Apiforol and luteoforol, which are produced by DFR. Although *CHS* and *DFR* are upregulated in *fka1-1* pericarps, the unknown polymerizing enzyme responsible for forming the red phlobaphene pigments may be expressed at a low level or may not function in the B73 background.

Anthocyanins are also flavonoid pigments, but the key anthocyanin biosynthesis gene *A2* (Zm00001d014914) has very low expression levels in the pericarp of B73 and *fka1-1* in RNA-seq data, which may be one of the significant reasons for *fka1-1* lacking obvious color.

FPKM values of *A2* in pericarp of B73 and *fka1-1* at 12, 20 and 30 DAP.

	B73-1	B73-2	B73-3	fka1-1	fka1-2	fka1-3
12 DAP	2.23	1.66	1.78	2.97	1.81	2.19
20 DAP	0.28	0.21	0.23	0.65	0.34	0.42
30 DAP	0.00	0.00	0.00	0.00	0.00	0.00

Some comments for the authors, not for this submission but for future work, follow. With regard to their metabolic flux hypothesis that an enhanced phenylpropanoid pathway shifts away from the production of auxin, would a dominant negative mutation of chalcone synthetase, namely *c2-ldf*, reverse the effect of the mutant? On the other hand, would the *r1-cherry* allele that produces extremely high levels of anthocyanin in the pericarp reverse the dent phenotype? Several accessions of *r1-cherry* in the reviewer’s collection all

show a flint phenotype but this does not necessarily mean anything. A test of whether *r1-cherry* can convert B73 to flint would be a valid test.

Response: Good idea!

We are also investigating whether the suppression of the chalcone synthetase gene *C2* can reverse the effect of *fka1-1*. We have developed *CHSRNAi* lines in the B73 background and are currently introgressing *fka1-1* into these *CHSRNAi* lines. We are eagerly anticipating the results.

Additionally, we find it intriguing that your *r1-cherry* mutants, displaying high levels of anthocyanin in the pericarp, exhibit the flint phenotype. B73 lacks color. By overexpressing *A2* or *R1* in B73, we can test whether elevated levels of anthocyanin can transform B73 into a flint variety.

Thanks again for your suggestions and comments.

Reviewer #2 (Remarks to the Author):

What a fantastic manuscript! This study identifies a novel genetic determinant of flint kernel architecture and provides an exhaustive set of data, from molecular to genetic, establishing a role for an auxin response factor (ARFTF17) in maize kernel development. This manuscript was a joy to read and is sure to be highly-cited. The data was very high quality and very exciting.

Response: Thanks for your comments.

Minor comments for suggested edits:

1. Line 36: Indicate which PIN1 gene(s) are regulated by ARFTF17 (there are several in maize).

Response: Thanks for your suggestion. The gene name is consistent with that in MaizeGDB. We also included its gene model association (Zm00001d044812) in the Results section when it was first mentioned.

2. Lines 99 & 105: Suggest re-wording for clarity regarding the nature of the ARFTF17 mutation (missense, nonsense, knock-down, null, etc). Are the phenotypes due to altered ARFTF17 function, structure, or expression level?

Response: Thanks for your suggestion. Two EMS mutants *fka1-1* and *fka1-2*, and a CRISPR line *fka1-3* are all null mutants, showing the same flint kernel architecture. They all contain a premature stop codon (Supplementary Fig.2j and Figure 2d).

3. Data S4 – add non-ambiguous gene identifiers for ARFTF17 and PIN1 (there are several PIN1 genes in maize and ARF annotation is variable depending on which phylogeny and genome annotation version is being referenced).

Response: Thanks for your suggestion. We added gene model associations of ARFTF17 and PIN1 in Data S4.

4. Line 152-153 should be moved to the discussion.

Response: Thanks for your suggestion. Although this sentence could be relocated to the discussion section, it is included here to enhance the understanding of crown architecture formation in B73 and *fka1-1*.

5. Lines 468-469: could the authors speculate on that contributes to apical patterning in the kernel during development? For example, could this pattern be established due to a hormone gradient (such as auxin) or differential

expression of metabolic genes or physical properties of the kernel? Does the integument play a role in setting up the proximal-distal pattern?

Response: Thank you for the suggestions. It is possible that auxin gradients, differential expression of metabolic genes, or physical properties of kernels may influence the formation of dent and flint kernel, but we do not have data to suggest this. However, these ideas provide new directions for our future research on the formation of dent and flint maize in natural populations. This is discussed in this section.

6. Line 491-492: Suggest changing the phrase “indicating ARFTF17 cannot directly regulate its downstream target genes” to “suggesting that additional factor(s) may be required for ARFTF17 DNA binding properties.”

Response: Thanks for your suggestion. This has been revised.

7. Line 492: change “needs to interact” to “can interact”.

Response: Thanks for your suggestion. This has been revised.

8. Line 550: change the word “study” to “investigation”.

Response: Thanks for your suggestion. This has been revised.

9. Add scale bars to all micrographs in figures (i.e. Figure 1a,b,f; Fig 2c,e; Fig 3a,b,e,f; Fig 4a,d,e,i,j; Fig 5g,h; Fig 6b,c;) and extended figures.

Response: Thanks for your suggestion. These have been revised.

10. Line 1130: Indicate amount of cDNA template used in each RT-qPCR.

Response: Thanks for your suggestion. About 80-100 μ g cDNA of each sample was used for RT-qPCR.

11. Line 1183-184: Indicate the excitation and emission wavelengths used for imaging GFP.

Response: Thanks for your suggestion. The sentence was revised as: The GFP fluorescent signal was observed with 488-nm excitation and 510-nm emission under an LSM880 confocal microscope (Zeiss, Jena, Germany).